# TPRU: Advancing Temporal and Procedural Understanding in Large Multimodal Models

**Zhenkun Gao**[1,2], **Xuhong Wang**[2*], **Xin Tan**[1,2*], **Yuan Xie**[1,3]

[1]East China Normal University, Shanghai, China
[2]Shanghai Artificial Intelligence Laboratory, Shanghai, China
[3]Shanghai Innovation Institute, Shanghai, China
`51275901149@stu.ecnu.edu.cn, wangxuhong@pjlab.org.cn,`
`{xtan, yxie}@cs.ecnu.edu.cn`

## Abstract

Multimodal Large Language Models (MLLMs), particularly smaller, deployable variants, exhibit a critical deficiency in understanding temporal and procedural visual data, a bottleneck hindering their application in real-world embodied AI. This gap is largely caused by a systemic failure in training paradigms, which lack large-scale, procedurally coherent data. To address this problem, we introduce TPRU, a large-scale dataset sourced from diverse embodied scenarios such as robotic manipulation and GUI navigation. TPRU is systematically designed to cultivate temporal reasoning through three complementary tasks: Temporal Reordering, Next-Frame Prediction, and Previous-Frame Review. A key feature is the inclusion of challenging negative samples, compelling models to transition from passive observation to active, cross-modal validation. We leverage TPRU with a reinforcement learning (RL) fine-tuning methodology, specifically targeting the enhancement of resource-efficient models. Experiments show our approach yields dramatic gains: on our manually curated TPRU-Test, the accuracy of TPRU-7B soars from 50.33% to 75.70%, a state-of-the-art result that significantly outperforms vastly larger baselines, including GPT-4o. Crucially, these capabilities generalize effectively, demonstrating substantial improvements on established benchmarks. The codebase is available at `https://github.com/Stephen-gzk/TPRU/`.

## 1 Introduction

Multimodal Large Language Models (MLLMs) have demonstrated impressive capabilities (Jin et al., 2025), with leading large-scale open-source (Bai et al., 2025; Zhu et al., 2025) and proprietary models (Hurst et al., 2024) achieving remarkable performance on a wide range of vision-language tasks. However, this progress masks a critical and widening gap: while massive, expensive models show emerging competence, their smaller and more efficient counterparts struggle profoundly with complex reasoning. Especially when they try to understand temporal and procedural image sequences (Song et al., 2025; Tang et al., 2025; Zhang et al., 2025). This capability gap is not merely an academic concern but a primary obstacle hindering the deployment of MLLMs in real-world and interactive applications. Downstream tasks like robotic manipulation, embodied navigation (Li et al., 2025b; Gu et al., 2025), and instruction following often operate on resource-constrained edge devices where deploying dozens or hundreds billion parameter models is infeasible (Ji et al., 2025; Lu et al., 2024; Savva et al., 2019). Consequently, the inability of small models to grasp state changes and procedural logic represents a fundamental bottleneck for the entire field of embodied AI.

The root of this deficiency lies not in model scale alone, but in a systemic failure of the prevailing training paradigm. Existing paradigms predominantly focus on aligning text with a single image (Li et al., 2023) or treating multiple images as an unordered collection (Jiang et al., 2024). Although datasets like LLaVA-NeXT-Interleave (Li et al., 2024) incorporate multi-frame inputs derived from videos, they primarily emphasize general sequential content comprehension rather than the fine-grained temporal and procedural understanding. This approach overlooks the critical distinction

---
*Corresponding authors.

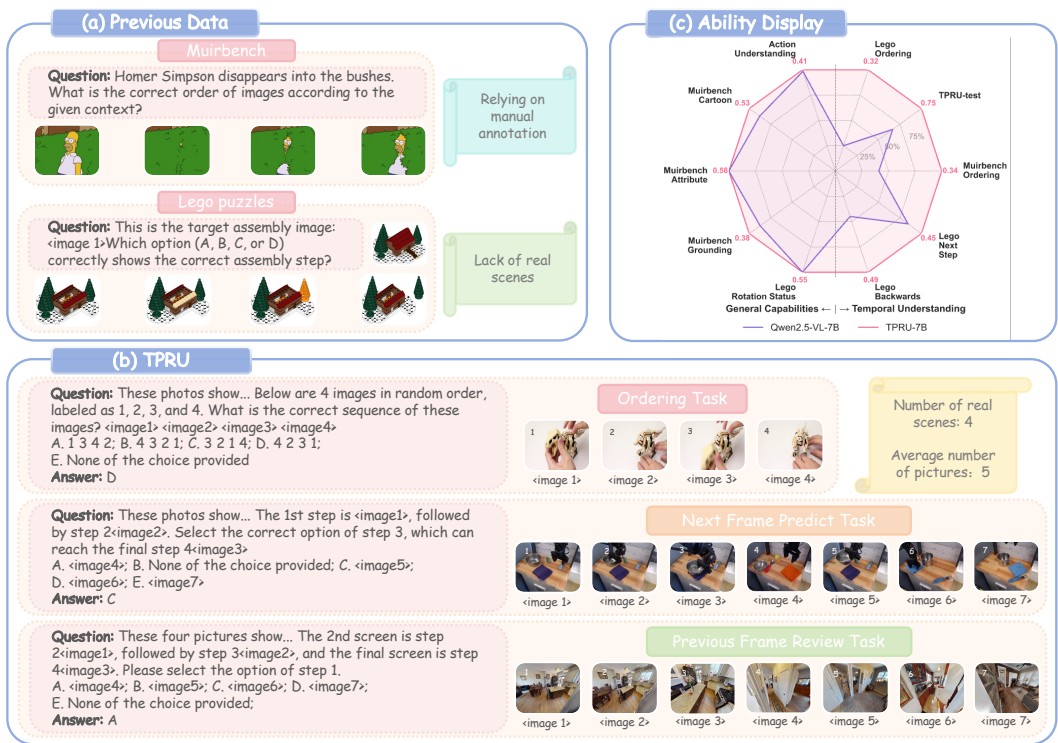

Figure 1: An overview of our TPRU dataset. Unlike prior synthetic datasets (a), TPRU is built from real-world scenarios and structured into temporal tasks (b). As shown in the ability display (c), TPRU-7B achieves significant performance gains in temporal understanding.

between understanding a set of images and comprehending a sequence of images. As shown in Figure 1(a), the community's response has been to create evaluation-only benchmarks that repeatedly confirm this failure (Wang et al., 2024; Tang et al., 2025), rather than addressing the root cause, which is the absence of large-scale, high-quality real-world sequential data for training. This oversight stems from the inherent difficulty of capturing the complex, continuous transformations of real-world actions. As a result, smaller models are evaluated on a sophisticated skill they were never systematically taught, leading to poor performance in detecting procedural errors and leaving genuine procedural understanding an unsolved problem for deployable AI systems (Song et al., 2025; Fu et al., 2024a; Tan et al., 2025; Wang et al., 2026).

To bridge this critical capability gap, particularly for resource-constrained models, we introduce TPRU (Temporal-Procedural Understanding dataset), a novel dataset designed to bridge the gap between evaluation and training. First, to address data scarcity and authenticity, TPRU provides a large-scale training set (24,750 QA pairs, 126,000 images) sourced from four diverse and authentic embodied scenarios: robotic manipulation, embodied navigation, mobile GUI interaction, and LEGO assembly. More importantly, as depicted in Figure 1(b), TPRU is not just a data collection but a systematically structured dataset designed to cultivate a deep procedural understanding through three complementary task formats: **Temporal Reordering**, **Next-Frame Prediction** and **Previous-Frame Review**. To ensure models develop true comprehension beyond superficial heuristics, TPRU incorporates a significant number of challenging negative samples with deliberate inconsistencies, forcing models to transition from passive "seeing" to active validation. To benchmark this capability, we also present the TPRU-Test, a manually curated set of 461 challenging instances for rigorous evaluation.

Leveraging our TPRU dataset, we employed a reinforcement learning (RL) strategy to fine-tune a suite of Qwen2.5-VL models, focusing on smaller parameter counts. The results are striking. Our RL-finetuned models not only show massive improvements over their base versions but also significantly outperform existing state-of-the-art (SOTA) MLLMs on our proposed TPRU test set.

Remarkably, our TPRU-7B and TPRU-32B surpass the performance of the much larger proprietary model GPT-4o and large-scale open-source models. Furthermore, as shown in Figure 1(c), TPRU-7B exhibits substantial gains on established multi-image benchmarks like MuirBench (Wang et al., 2024) and LEGO-Puzzles (Tang et al., 2025). These findings demonstrate that the temporal reasoning gap in small models is not an inherent limitation of their scale but a solvable challenge of targeted data and training. We have unlocked SOTA-level procedural understanding in models small enough for practical, real-world deployment. Our contributions are summarized as below:

- We construct TPRU, a new, large-scale, high-quality multi-image dataset focused on fine-grained temporal and procedural understanding, designed to empower smaller models for embodied contexts. The dataset and its creation methodology will be publicly released.

- We present a challenging held-out test set, TPRU-Test, manually curated and verified to rigorously evaluate temporal understanding in MLLMs.

- We propose and validate an effective reinforcement learning-based training methodology that enables small-to-medium-sized MLLMs to achieve and even surpass the temporal understanding capabilities of vastly larger models. Extensive experiments demonstrate this superiority and strong generalization on both our TPRU-test and existing public benchmarks.

## 2 RELATED WORK

While Multimodal Large Language Models excel at single-image comprehension (Fu et al., 2024a; Li et al., 2025a; Zhang et al., 2024a), multi-image reasoning remains a challenge. Recent work has thus expanded beyond the single-frame paradigm to tackle complex real-world tasks (Wang et al., 2025b), necessitating specialized instruction-tuning datasets and evaluation benchmarks.

### 2.1 MULTI-IMAGE TRAINING DATA

To enhance the multi-image capabilities of Multimodal Large Language Models (MLLMs), researchers have constructed large-scale instruction-tuning datasets. For instance, Mantis-Instruct (Jiang et al., 2024) adopts a skill-oriented strategy, efficiently imbuing models with four core abilities via a meticulously constructed 721K-sample dataset. LLaVA-NeXT-Interleave (Li et al., 2024) advances this direction by leveraging its 1.18M-sample M4-Instruct dataset and a unified interleaved data format to seamlessly handle multi-image, video, and 3D scenarios. Addressing conversational depth, MMDU (Liu et al., 2024) and MMCR (Yan et al., 2025) provide large-scale datasets specifically for training models on coherent reasoning in multi-turn dialogues. Furthermore, datasets have expanded into specific domains, such as RoboBrain's ShareRobot (Ji et al., 2025), which provides planning and affordance annotations for robotics tasks, and GUI Odyssey (Lu et al., 2024), which focuses on cross-application mobile device navigation (Luo et al., 2025). In parallel, innovative data generation paradigms are emerging that move beyond reliance on manual annotation. Jigsaw-R1 (Wang et al., 2025c), for example, enhances models' spatial awareness by programmatically generating jigsaw puzzle tasks for rule-based visual reinforcement learning. Taking this further, MiCo (Chen et al., 2025) proposes a fully self-supervised reinforcement learning framework, enabling models to learn complex reasoning from programmatically constructed contrastive image triplets without any instruction data.

While these datasets have significantly advanced multi-image instruction tuning, they often treat images as an unordered collection. Even sequential datasets lack a systematic framework for teaching procedural flow principles. Our work addresses this gap by introducing TPRU. Through its complementary three temporal tasks, TPRU is specifically engineered to instill a foundational understanding of procedural dynamics.

### 2.2 BENCHMARKS FOR MULTI-IMAGE EVALUATION

Concurrently with the development of training data, a suite of rigorous benchmarks has been established to evaluate these emerging capabilities. For comprehensive and robust evaluation, Muir-Bench (Wang et al., 2024) presents a 2,600-question test whose key innovation is a pairwise design—matching each answerable question with a minimally different, unanswerable variant to rig-

orously test against hallucination. Targeting specific reasoning domains, STRIPCIPHER (Wang et al., 2025b) leverages wordless comic strips to assess narrative and temporal logic, while TempVS (Song et al., 2025) focuses on event ordering with a design that resists single-modality shortcuts. For spatial and physical reasoning, LEGO-Puzzles (Tang et al., 2025) creates a challenging testbed for multi-step planning based on LEGO instructions, and MV-Math (Wang et al., 2025a) fills a critical gap in multi-visual context mathematical reasoning using real K-12 educational materials. Other benchmarks probe more fundamental visual abilities; BLINK (Fu et al., 2024b) aims to decouple core visual perception from linguistic reasoning, and MMRA (Wu et al., 2024) evaluates the ability to identify cross-image relations at multiple granularities. Finally, to systematize the application of these benchmarks, toolkits like VLMEvalKit (Duan et al., 2024) provide a standardized evaluation framework, greatly facilitating reproducible and comprehensive assessment across the community.

A critical limitation of existing work is their evaluation-only nature, creating a gap between training and testing. TPRU bridges this gap by providing a large-scale structured training set alongside a challenging test set, unifying the development and evaluation of procedural understanding.

## 3 TPRU

While current Multi-modal Large Language Models excel on static single-image tasks, their performance degrades sharply as the number of input images increases (Li et al., 2024). This deficiency is exacerbated when processing image sequences that represent a coherent process or event. Recent work shows existing MLLMs largely fail to comprehend temporal dynamics and sequential relationships between visual frames. This limitation is starkly revealed on benchmarks for narrative comics, procedural instructions, and event ordering (Song et al., 2025; Tang et al., 2025; Wang et al., 2025b; 2024). Consequently, the inability to grasp temporal dynamics severely hinders their applicability in real-world scenarios that demand comprehension of procedural activities and evolving states (Tang et al., 2025; Ji et al., 2025).

To systematically enhance and evaluate the capability of MLLMs in comprehending image sequences with temporal and procedural order, we propose the TPRU dataset. The dataset consists of two components. TPRU-25k is a fine-tuning set with 24,750 samples across four procedural scenarios, designed to enhance the model's temporal and procedural understanding. TPRU-test is an evaluation benchmark comprising 461 manually annotated samples across five application scenarios. The detailed data sourcing and construction methodologies for TPRU-25k and TPRU-test are elaborated in Sections 3.1 and 3.2, respectively.

### 3.1 TPRU-25K

**Data Sources.** TPRU aims to provide MLLMs with high-quality, multi-image sequential data characterized by clear procedural and temporal structures. To construct a dataset with coherent procedural logic and ensure that the image sequences represent meaningful, ordered events, we sourced data from four diverse and complementary real-world scenarios: (1) **Robotic Manipulation.** Data is primarily derived from the "planning" tasks in the ShareRobot dataset (Ji et al., 2025), where we sample video frames to create discrete action sequences. (2) **LEGO Assembly.** Data is curated from 36 high-quality stop-motion videos from the YouTube creator Arvin Bricks, providing blur-free, state-distinct images ideal for part-to-whole reasoning. (3) **GUI Operation.** A novel dataset constructed from four-step screenshot sequences from GUI Odyssey (Lu et al., 2024) to capture goal-driven digital workflows. (4) **Embodied Navigation.** This category consists of ordered visual observations from agents navigating in simulated environments like Habitat (Savva et al., 2019). The diversity of these sources ensures the data is not confined to a single domain, fostering the development of generalizable sequential understanding.

**Generation Pipeline** Our data generation pipeline systematically transforms raw sequential data from diverse sources into structured training instances. The process involves three main stages: sequence filtering, text description generation, and task formulation, as illustrated in Figure 2.

**(a) Filtering and Quality Control.** Our dataset is constructed from heterogeneous sources, including temporally sampled video frames and ordered screenshots from embodied agent tasks and mobile UI interactions. We process these diverse inputs into a canonical format of coherent image sequences, each containing three to four images. To ensure high data quality and procedural

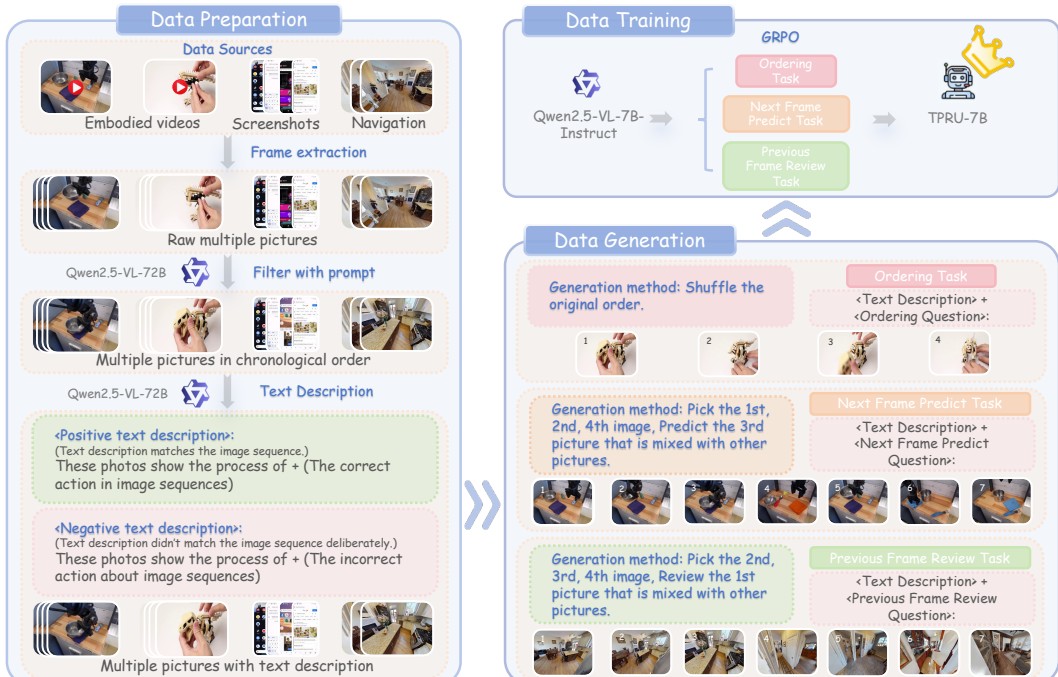

Figure 2: The TPRU dataset construction and training pipeline. Chronological image sequences from embodied sources are curated with both positive and negative text descriptions. These image sequences are then formulated into three tasks (Ordering, Next Frame Prediction, and Previous Frame Review) to fine-tune MLLMs for enhanced temporal and procedural understanding.

integrity, every image sequence is subjected to a rigorous filtering pipeline. We employ Qwen2.5-VL-72B (Bai et al., 2025) as an automated quality assessor to discard sequences exhibiting visual blurriness, abrupt scene transitions, or a lack of discernible temporal progression. This stringent protocol guarantees that only high-fidelity, logically coherent sequences are used for subsequent processing.

**(b) Description Generation and Robustness Enhancement.** For each filtered image sequence, we generate a corresponding textual description using Qwen2.5-VL-72B to reinforce the model's core vision-language alignment. To specifically bolster robustness and mitigate hallucination, we also introduce a negative sampling strategy. This involves creating a subset of instances where the textual description is deliberately mismatched with the visual content (e.g., pairing the instruction "pick up the fork" with images of "putting down a knife"). For these challenging cases, the target output is formulated as "None of the choices provided" (Wang et al., 2024). This forces the model to perform explicit cross-modal verification rather than relying solely on textual priors.

**(c) Task Formulation.** Based on the curated image sequences and their corresponding textual descriptions, we formulate three distinct yet complementary tasks designed to comprehensively enhance the model's temporal and procedural understanding ability.

- **Temporal Ordering.** The primary objective of this task is to evaluate and enhance the model's comprehension of an entire procedural timeline. We formulate this as a reordering problem. For a given image group, we shuffle the temporal order of the frames and provide the corresponding textual description. The model is then required to output the correct permutation that restores the reasonable sequence of the event.

- **Next Frame Prediction.** This task improves the model's grasp of temporal coherence and procedural flow. The model is presented with the initial, second, and terminal frames of a four-frame procedural sequence and must select the correct intervening third frame from a set of candidates. These candidates include distractors from other similar scenarios. This task directly simulates the immediate planning required for robotic agents to anticipate the direct consequences of an action.

- **Previous Frame Review.** This task enhances the model's ability to reconstruct the historical context of a procedural segment. The model is given the final three frames of a four-frame sequence and is tasked with identifying the correct initial frame from a set of candidates. This capability improves the model's understanding of procedural prerequisites and its ability to trace an observed event back to its origin, a fundamental aspect of comprehensive temporal understanding.

Collectively, these three complementary tasks are engineered to advance MLLMs beyond static image analysis. By jointly addressing temporal ordering, forward prediction, and backward review, our approach endows the model with a more profound and structured comprehension of procedural dynamics, significantly enhancing its ability to interpret the temporal evolution of events depicted in image sequences.

## 3.2 TPRU-TEST

To rigorously evaluate the temporal and procedural understanding capabilities of MLLMs, we introduce TPRU-Test, a dedicated, held-out evaluation set. TPRU-Test is meticulously curated and verified by human experts to ensure high quality and present novel generalization challenges. Its composition is deliberately diverse, incorporating the most demanding instances from our four core domains (Robotic Manipulation, LEGO Assembly, GUI Operation, and Embodied Navigation) alongside complex, human activities from the EPIC-KITCHENS (Damen et al., 2020) dataset to probe model robustness. TPRU-Test inherits three complementary tasks of our training set, assessing temporal ordering, next-frame prediction, and previous-frame review. The curation protocol was exceptionally stringent. Each instance underwent a multi-stage human review where annotators picked and verified the ground-truth image sequence, authored plausible yet incorrect distractors, and ensured question clarity. Subsequently, every instance was cross-verified by at least one other expert to eliminate errors and subjective judgments. This process yields a high-quality benchmark of 461 instances across 5 distinct scenarios, engineered to provide a robust measure of genuine progress in temporal and procedural understanding for MLLMs.

## 4 EXPERIMENTS

In this section, we conduct comprehensive experiments to validate the effectiveness of our proposed TPRU dataset. We fine-tuned Qwen2.5-VL on TPRU-25k and evaluated on TPRU-test as well as on established public benchmarks. In Section 4.1, we assess the model's performance on MuirBench (Wang et al., 2024) and LEGO-Puzzles (Tang et al., 2025) benchmarks which include tasks relevant to temporal and procedural understanding. And we present the primary results on our TPRU-test set in Section 4.2. To ensure our method does not degrade performance on broader tasks, we evaluate on general-purpose benchmarks in Section 4.3. Finally, in Section 4.4, we report a series of ablation studies to investigate the contribution of key components of our dataset. The hyperparameter Settings and reward designs of the experiment can be obtained respectively in Appendix A and F.

### 4.1 EVALUATION OF TEMPORAL RELATED BENCHMARKS

**Performance on MuirBench.** As presented in Table 1, our TPRU-finetuned models demonstrate significant improvements over their Qwen2.5-VL base models across all scales on the MuirBench (Wang et al., 2024). Notably, our TPRU-32B model achieves an overall accuracy of 68.42%, outperforming the powerful proprietary model GPT-4o (68.00%) and closely matching the much larger Qwen2.5-VL-72B.

The most substantial gains are observed in the Ordering sub-task, which directly aligns with the temporal reasoning skills targeted by our TPRU dataset. TPRU-32B achieves a remarkable 45.31% in this category, drastically surpassing both its base model and GPT-4o. This trend is consistent across scales, with the score of TPRU-7B also more than doubling from 14.06% to 34.38%. Beyond temporal tasks, our training methodology enhances broader relational reasoning abilities, evidenced by significant improvements in tasks such as Difference Spotting and Visual Retrieval. These results confirm that our approach not only instills specialized temporal skills but also strengthens general cross-image reasoning capabilities.

Table 1: **Performance on MuirBench.** The light gray rows show the absolute improvement (in percentage points) of our models over their corresponding Qwen2.5-VL base models. Gains are shown in red, and losses in blue.

| Model | Action Underst. | Attribute Similarity | Cartoon Underst. | Counting | Diagram Underst. | Difference Spotting | Geographic Underst. | Image-Text Matching | Ordering | Scene Underst. | Visual Grounding | Visual Retrieval | Overall |
|---|---|---|---|---|---|---|---|---|---|---|---|---|---|
| *Open-source* | | | | | | | | | | | | | |
| InternVL3-78B | 48.17 | 61.73 | 44.87 | 50.85 | 83.17 | 55.59 | 60.00 | 79.31 | **31.25** | 69.35 | 44.05 | 66.10 | 64.65 |
| InternVL3-38B | 44.51 | 66.33 | 46.15 | 45.30 | 78.14 | 61.18 | 63.00 | 77.80 | **32.81** | 61.29 | 36.90 | 72.95 | 64.12 |
| Qwen2.5-VL-72B | 50.00 | 59.18 | 42.31 | 49.57 | 89.45 | 60.59 | 50.00 | 87.93 | **40.63** | 76.34 | 46.43 | 78.42 | **69.35** |
| Qwen2.5-VL-32B | 36.59 | 51.02 | 47.44 | 45.30 | 82.41 | 58.53 | 47.00 | 85.78 | **26.56** | 72.04 | 41.67 | 67.12 | 63.73 |
| Qwen2.5-VL-7B | 40.85 | 58.67 | 46.15 | 34.19 | 77.89 | 54.41 | 49.00 | 72.63 | **14.06** | 61.83 | 33.33 | 63.70 | 58.35 |
| Qwen2.5-VL-3B | 36.59 | 44.39 | 46.15 | 32.91 | 58.04 | 46.47 | 49.00 | 54.09 | **9.38** | 59.14 | 40.48 | 38.70 | 46.62 |
| *Proprietary* | | | | | | | | | | | | | |
| Gemini-2.5-Flash | 50.00 | 49.49 | 60.26 | 82.05 | 92.71 | 70.00 | 47.00 | 86.64 | **46.88** | 83.87 | 59.52 | 70.89 | **73.73** |
| GPT-4o | 44.51 | 56.12 | 51.28 | 49.15 | 88.69 | 60.29 | 56.00 | 86.85 | **23.44** | 71.51 | 36.90 | 80.14 | 68.00 |
| Claude-3.5-Sonnet | 35.37 | 55.10 | 44.87 | 35.90 | 76.38 | 54.12 | 41.00 | 77.59 | **25.00** | 54.84 | 47.62 | 57.53 | 57.69 |
| *Ours* | | | | | | | | | | | | | |
| *TPRU-32B* | 40.24 | 51.02 | 51.28 | 47.44 | 85.68 | 63.24 | 60.00 | 87.28 | **45.31** | 75.81 | 44.05 | 80.14 | **68.42** |
| *Improvement* | +3.65 | 0.00 | +3.84 | +2.14 | +3.27 | +4.71 | +13.00 | +1.50 | **+18.75** | +3.77 | +2.38 | +13.02 | +4.69 |
| *TPRU-7B* | 41.46 | 57.65 | 47.44 | 34.62 | 82.91 | 63.82 | 63.00 | 82.11 | **34.38** | 67.74 | 38.10 | 75.68 | **65.04** |
| *Improvement* | +0.61 | -1.02 | +1.29 | +0.43 | +5.02 | +9.41 | +14.00 | +9.48 | **+20.32** | +5.91 | +4.77 | +11.98 | +6.69 |
| *TPRU-3B* | 39.63 | 51.02 | 47.44 | 40.17 | 67.34 | 55.88 | 84.00 | 68.32 | **23.44** | 72.04 | 53.57 | 66.10 | **59.31** |
| *Improvement* | +3.04 | +6.63 | +1.29 | +7.26 | +9.30 | +9.41 | +35.00 | +14.23 | **+14.06** | +12.90 | +13.09 | +27.40 | +12.69 |

**Performance on LEGO-Puzzles.** The efficacy of our approach is further validated on the LEGO-Puzzles benchmark, which directly evaluates procedural assembly reasoning as detailed in Table 2. Our models consistently outperform their base versions, with TPRU-7B achieving an overall score of 42.8%. This result confirms that the skills learned from our TPRU-25k dataset exhibit strong positive generalization to complex, structured assembly tasks. The most pronounced improvements are concentrated in the Multi-Step Reasoning category, confirming the targeted impact of our training methodology. For the TPRU-7B, performance on the Ordering task quadruples, soaring from 8.0% to 32.0%. Similarly, its capability in Backwards reasoning more than doubles, jumping from 22.0% to 49.0%, and a significant gain is also observed in Next-Step sub-task. These enhancements strongly indicate that the procedural and causal reasoning abilities cultivated by the TPRU dataset effectively transfer to the logical, sequential challenges inherent in the LEGO-Puzzles.

Table 2: **Performance on LEGO-Puzzles.** The light gray rows show the absolute improvement (in percentage points) of our models over their corresponding Qwen2.5-VL base models. Gains are shown in red, and losses in blue.

| Models | Height | Adjacency | Rotation | Multiview | Next-Step | Dependency | Rotation Stat. | Position | Backwards | Ordering | Outlier | Overall |
|---|---|---|---|---|---|---|---|---|---|---|---|---|
| *Open-source* | | | | | | | | | | | | |
| InternVL3-78B | 52.0 | 63.0 | 36.0 | 54.0 | 64.0 | 80.0 | 58.0 | 29.0 | 25.0 | **22.0** | 37.0 | 47.3 |
| InternVL3-38B | 40.0 | 60.0 | 39.0 | 55.0 | 57.0 | 81.0 | 59.0 | 33.0 | 47.0 | **12.0** | 36.0 | 47.2 |
| Qwen2.5-VL-72B | 43.0 | 58.0 | 38.0 | 39.0 | 57.0 | 76.0 | 57.0 | 52.0 | 74.0 | **43.0** | 43.0 | 52.7 |
| Qwen2.5-VL-32B | 35.0 | 60.0 | 38.0 | 52.0 | 45.0 | 79.0 | 51.0 | 45.0 | 66.0 | **43.0** | 43.0 | 50.6 |
| Qwen2.5-VL-7B | 20.0 | 57.0 | 32.0 | 47.0 | 38.0 | 67.0 | 56.0 | 29.0 | 22.0 | **8.0** | 25.0 | 36.5 |
| Qwen2.5-VL-3B | 29.0 | 55.0 | 30.0 | 36.0 | 32.0 | 65.0 | 48.0 | 19.0 | 16.0 | **4.0** | 25.0 | 32.6 |
| *Proprietary* | | | | | | | | | | | | |
| Gemini-2.5-Flash | 52.0 | 58.0 | 37.0 | 55.0 | 58.0 | 74.0 | 53.0 | 49.0 | 40.0 | **46.0** | 29.0 | 50.1 |
| GPT-4o | 49.0 | 66.0 | 41.0 | 51.0 | 65.0 | 87.0 | 51.0 | 51.0 | 53.0 | **72.0** | 49.0 | **57.7** |
| Claude-3.5-Sonnet | 39.0 | 60.0 | 42.0 | 48.0 | 61.0 | 78.0 | 58.0 | 37.0 | 49.0 | **54.0** | 64.0 | 53.6 |
| *Ours* | | | | | | | | | | | | |
| *TPRU-32B* | 34.0 | 61.0 | 35.0 | 47.0 | 55.0 | 76.0 | 52.0 | 48.0 | 70.0 | **49.0** | 48.0 | **52.3** |
| *Improvement* | -1.0 | +1.0 | -3.0 | -5.0 | +10.0 | -3.0 | +1.0 | +3.0 | +4.0 | **+6.0** | +5.0 | +1.7 |
| *TPRU-7B* | 23.0 | 56.0 | 37.0 | 40.0 | 45.0 | 67.0 | 55.0 | 31.0 | 49.0 | **32.0** | 36.0 | **42.8** |
| *Improvement* | +3.0 | -1.0 | +5.0 | -7.0 | +7.0 | 0.0 | -1.0 | +2.0 | +27.0 | **+24.0** | +11.0 | +6.3 |
| *TPRU-3B* | 30.0 | 54.0 | 32.0 | 40.0 | 37.0 | 69.0 | 49.0 | 14.0 | 30.0 | **8.0** | 24.0 | 35.2 |
| *Improvement* | +1.0 | -1.0 | +2.0 | +4.0 | +5.0 | +4.0 | +1.0 | -5.0 | +14.0 | **+4.0** | -1.0 | +2.6 |

## 4.2 EVALUATION OF GENERAL BENCHMARKS

To ensure our specialized training does not degrade general capabilities, we evaluated our model on a range of broad multi-image benchmarks, as shown in Table 3. The results confirm that our finetuned models maintain or slightly improves performance across these diverse tasks. This pattern of stable to positive gains, such as on MMMU (+2.6) and MMCR (+1.08) for TPRU-7B, demonstrates that our

method for enhancing temporal reasoning successfully avoids catastrophic forgetting and preserves the model's foundational abilities. The relevant content of these benchmarks can be obtained from the appendix.

Table 3: **Evaluation on general multi-image benchmarks.** Results show that our TPRU models maintain comparable performance to their base models, indicating that our specialized training does not degrade general capabilities.

| Model | MME-RealWorld-Lite | BLINK | RealWorld QA | MMCR | MMTBench | MMStar | MMMU-Dev | Overall |
|---|---|---|---|---|---|---|---|---|
| *Open-source* | | | | | | | | |
| InternVL3-78B | 65.40 | 66.30 | 78.00 | 20.29 | 73.20 | 72.50 | 64.20 | 62.84 |
| InternVL3-38B | 67.30 | 64.49 | 75.60 | 21.38 | 71.80 | 71.50 | 62.00 | 62.01 |
| Qwen2.5-VL-32B | 45.96 | 59.34 | 68.89 | 37.32 | 58.89 | 54.93 | 34.67 | 51.43 |
| Qwen2.5-VL-7B | 44.55 | 54.76 | 68.10 | 22.83 | 61.46 | 61.67 | 44.40 | 51.11 |
| Qwen2.5-VL-3B | 41.94 | 48.97 | 65.36 | 19.20 | 60.47 | 54.40 | 44.67 | 47.86 |
| *Ours* | | | | | | | | |
| *TPRU-32B* | 44.92 | 58.81 | 69.80 | 39.86 | 56.25 | 52.26 | 34.00 | 50.84 |
| *TPRU-7B* | 45.34 | 55.86 | 69.54 | 23.91 | 61.85 | 61.13 | 47.00 | 52.09 |
| *TPRU-3B* | 39.19 | 48.13 | 66.54 | 21.01 | 60.89 | 54.93 | 44.67 | 47.91 |

## 4.3 MAIN RESULTS ON TPRU-TEST

We evaluate the core efficacy of our approach on the proposed **TPRU-test**, a benchmark specifically designed to assess fine-grained temporal ordering, causal prediction, and procedural consistency. The results, presented in Figure 3, demonstrate that fine-tuning with our TPRU dataset with RL methodology yields substantial and consistent performance improvements across various model scales. Notably, the accuracy of TPRU-7B soars from 50.33% to **75.70%**, while the TPRU-3B shows a similarly strong improvement from 37.96% to **60.95%**. This level of performance is highly competitive: TPRU-7B surpasses the powerful proprietary model GPT-4o by a significant margin. These results unequivocally validate the effectiveness of our training paradigm. The dramatic performance gains underscore that targeted training on procedurally-grounded data, enriched with challenging negative samples and optimized via reinforcement learning, is a potent strategy for instilling robust sequential reasoning capabilities in MLLMs.

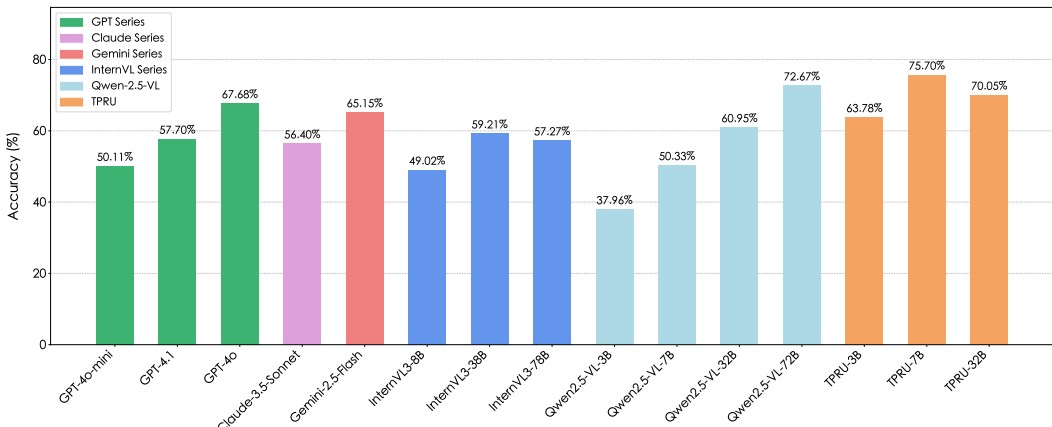

Figure 3: Performance of different models on TPRU-test.

## 4.4 ABLATION STUDIES

To systematically investigate the contributions of the core components of our methodology, we conducted a series of ablation studies. We evaluate the impact of task composition, negative samples and data volume. All experiments are conducted by training variants of the Qwen2.5-VL-7B model,

with performance evaluated on MuirBench and Lego-Puzzles. The detailed results are presented in Appendix.

Table 4: Ablation study on the effect of different TPRU data components for the **Qwen2.5-VL-7B** model on MuirBench and LEGO-Puzzles benchmarks. The inclusion of a component is marked with a ✓. All scores are Overall Accuracy (%). The volume of data remains consistent.

| TPRU Data Components | | | Benchmark Accuracy (%) | |
| --- | --- | --- | --- | --- |
| Ordering | Previous Frame Review | Next Frame Predict | **MuirBench** | **LEGO-Puzzles** |
| ✓ | | | 60.8 | 39.0 |
| | ✓ | | 61.7 | 38.0 |
| | | ✓ | 61.6 | 41.1 |
| ✓ | ✓ | | 62.5 | 40.3 |
| ✓ | | ✓ | 62.2 | 40.3 |
| | ✓ | ✓ | 62.2 | 39.1 |
| ✓ | ✓ | ✓ | **63.8** | **42.3** |

**Impact of Task Composition.** We conducted an ablation study on a small-scale dataset of 8,250 samples to analyze the individual and combined contributions of our core training tasks, with results presented in Table 4. The findings reveal a clear synergistic effect. While each task component individually improves baseline performance, their pairwise combinations yield further gains. A model trained on the complete dataset integrating all three tasks ultimately achieves the highest performance. This confirms that the diversity of these complementary reasoning skills is crucial for fostering a comprehensive and robust understanding of temporal and procedural logic.

**Efficacy of Negative Samples.** A core design choice in TPRU is the inclusion of negative samples, which are instances with deliberate procedural inconsistencies that force the model to reject all options. An ablation study confirms their impact. As illustrated in Figure 4a, training without these negative samples significantly degrades performance on temporal reasoning benchmarks like LEGO-Puzzles and MuirBench. This result validates our hypothesis that teaching a model to explicitly reject invalid logic is critical for advancing from pattern recognition to robust procedural understanding.

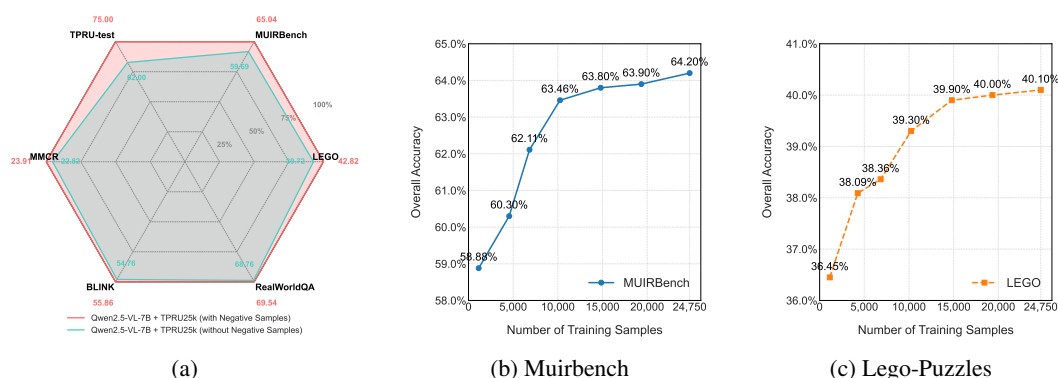

Figure 4: Ablation analysis. (a) Ablation on negative samples. (b) and (c) show the performance with different training samples.

**Impact of Data Volume.** We evaluated training on data subsets up to the full 24,750 samples. As shown in Figures 4b and 4c, performance on MuirBench and LEGO-Puzzles improves with data size but begins to plateau near the full set. This pattern of diminishing returns indicates our dataset is sufficiently comprehensive to instill robust temporal understanding without requiring further scaling.

**Impact of Training Strategy.** To validate the necessity of our Reinforcement Learning framework, we conducted a direct comparison between the proposed GRPO approach and standard Supervised Fine-Tuning (SFT). Both experiments were performed using the Qwen2.5-VL-7B backbone under consistent hyperparameters via the LLaMA-Factory framework Zheng et al. (2024).

As shown in Table 5, while SFT provides a solid performance baseline, GRPO consistently achieves superior results across all benchmarks. These results indicate that GRPO is more effective than

Table 5: Ablation study on training paradigms. We compare the performance of standard Supervised Fine-Tuning (SFT) against our GRPO approach using the Qwen2.5-VL-7B backbone.

| Training Paradigm | TPRU-Test | MuirBench | LEGO-Puzzles |
|---|---|---|---|
| Supervised Fine-Tuning (SFT) | 72.88 | 63.03 | 40.60 |
| **TPRU-7B (GRPO)** | **75.70** | **65.04** | **42.82** |

Table 6: Performance comparison with state-of-the-art Video LLMs on MuirBench, LEGO-Puzzles, and TPRU-Test datasets. The "Ordering" columns denote specific subtasks requiring precise sequential reasoning. Best results are highlighted in **bold**.

| Model | MuirBench | | LEGO-Puzzles | | TPRU-Test |
|---|---|---|---|---|---|
| | Overall | Ordering | Overall | Ordering | |
| LLaVA-Video-Qwen2-7B | 37.88 | 15.63 | 31.91 | 1.0 | 38.61 |
| LLaVA-Video-Qwen2-72B | 41.81 | 10.94 | 41.64 | 5.0 | 44.03 |
| SmolVLM2-256M-Video | 27.92 | 21.88 | 28.18 | 0.0 | 17.79 |
| SmolVLM2-500M-Video | 25.92 | 7.8 | 27.27 | 0.0 | 16.92 |
| Long-VITA-16K | 53.07 | 17.19 | 34.45 | 2.0 | 39.26 |
| Qwen2.5-Omni-7B | 59.11 | 18.75 | 36.45 | 12.0 | 46.85 |
| **TPRU-7B (Ours)** | **65.04** | **34.38** | **42.82** | **32.0** | **75.70** |

SFT for this domain, as it moves beyond simple pattern imitation to better cultivate the advanced reasoning capabilities required for complex procedural multi-modal tasks.

### 4.5 COMPARISON WITH VIDEO-TEMPORAL BASELINES

To further elucidate the distinction between procedural temporal understanding and general video understanding, we conducted a comprehensive evaluation using several state-of-the-art Video Large Language Models. We posit that while general video understanding typically focuses on recognizing what events occur, procedural multi-image understanding places a greater emphasis on inferring the precise sequence and consequential relationships between discrete actions.

We compared our TPRU-7B model against strong open-source video baselines, including LLaVA-Video (Zhang et al., 2024b), SmolVLM2 (Marafioti et al., 2025), Long-VITA Shen et al. (2025), and Qwen2.5-Omni (Xu et al., 2025), on the MuirBench, LEGO-Puzzles, and our TPRU-Test datasets. As presented in Table 6, TPRU-7B significantly outperforms these powerful video models across all three benchmarks.

Results reveal a critical gap in current video models regarding fine-grained temporal ordering. For instance, on the MuirBench Ordering subtask, the top-performing baseline Qwen2.5-Omni scores only 18.8, whereas TPRU-7B achieves 34.38.

These findings support our hypothesis: while general video pre-training provides fundamental temporal awareness, it is insufficient for precise procedural understanding. Our approach bridges this gap through targeted data synthesis and training designed for discrete state changes.

## 5 CONCLUSION

In this work, we addressed the critical failure of MLLMs in comprehending visual sequences by introducing TPRU, a large-scale dataset designed to systematically teach temporal and procedural logic. Our experiments show that fine-tuning with a reinforcement learning strategy on TPRU yields dramatic performance gains, with TPRU-7B not only dominating its baseline but also outperforming the much larger GPT-4o and generalizing strongly to benchmarks like MuirBench and LEGO-Puzzles. The primary contribution of this work is the demonstration that targeted, procedurally-grounded data can effectively close the reasoning gap for smaller, more efficient models, moving the frontier of capable AI from massive systems towards practical, deployable agents.

## ETHICAL CONDUCT AND SOCIETAL IMPACT STATEMENT

This research was conducted with a steadfast commitment to ethical integrity, in strict accordance with the ICLR Code of Ethics. All experimental procedures and data handling protocols comply with applicable national and international laws, institutional regulations, and established ethical standards.

The data utilized in this study were sourced exclusively from publicly available, open-access datasets, or were obtained with explicit and appropriate authorization from the data providers. To safeguard individual privacy and ensure data security, all datasets underwent rigorous preprocessing, including anonymization and the removal of personally identifiable information (PII) where applicable.

Specifically regarding the video content sourced from the "Arvin Bricks" YouTube channel, these materials are utilized strictly for non-commercial, transformative academic research. Consistent with practices in recent literature (Ju et al., 2025), we operate under the principle of fair use for publicly available creative works governed by the Standard YouTube License. Our processing and analysis of this data are conducted solely to advance scientific understanding, with no commercial intent or redistribution of the original assets.

The primary objective of this work is to contribute to the advancement of scientific knowledge. We have carefully considered the potential societal impacts of our research and have found no foreseeable risks of direct harm or misuse. The authors explicitly disavow any application of this work for malicious or unethical purposes. Furthermore, the authors declare that there are no competing financial or personal interests that could have influenced the outcomes of this research.

## REPRODUCIBILITY STATEMENT

To ensure the transparency and reproducibility of our research, we provide a detailed description of our methodology, datasets, and experimental setup within the main body and appendix of this paper. To facilitate verification and extension of our work by the research community, all associated source code, data preprocessing scripts, and necessary model files are publicly available. We have released the complete set of materials to foster collective progress in the academic community.

## ACKNOWLEDGEMENTS

This work was supported by the Science and Technology Commission of Shanghai Municipality (Grant Nos. 25511102700, 25511103300, 25511104302), the National Natural Science Foundation of China (Grant Nos. 62302167, U23A20343, W2521174), and the Young Elite Scientists Sponsorship Program by CAST (Grant No. YESS20240780).

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

## A EXPERIMENTAL SETUP

**Hyperparameters.** We fine-tuned the Qwen2.5-VL model (Bai et al., 2025) on our TPRU dataset. Our reinforcement learning methodology was implemented using the Easy-R1 framework (Zheng et al., 2025), employing the Group-wise Preference Optimization (GRPO) algorithm (Shao et al., 2024). Key settings included KL regularization with a coefficient of 0.01 and the generation of 5 rollouts per training sample. We used the AdamW optimizer with a learning rate of 1e-6 and trained for 2 epochs on a cluster of 8 NVIDIA A800 GPUs.

**Evaluation Benchmarks.**

To comprehensively evaluate the performance of our model, we assessed its capabilities on a wide variety of benchmarks. To test its generalized multi-image and multimodal reasoning abilities, we selected several established public benchmarks, including MME-RealWorld-Lite (Zhang et al., 2024a) and RealWorldQA (xAI team, 2024), BLINK (Fu et al., 2024b), MMCR (Yan et al., 2025) and MMStar (Chen et al., 2024), MMTBench (Ying et al., 2024), MMMU (Yue et al., 2024), Muir-Bench (Wang et al., 2024), and LEGO-Puzzles (Tang et al., 2025). Furthermore, to specifically measure the improvements in temporal and procedural understanding, the core focus of our work. We evaluated our model on our proposed TPRU-test. To ensure a standardized and reproducible evaluation process across all benchmarks, we utilized the open-source VLMEvalKit framework (Duan et al., 2024). We also integrated our TPRU-test into this framework, allowing for a consistent and streamlined evaluation methodology for both existing and our newly proposed tasks.

## B PROMPTS FOR MICRO IMAGE SEQUENCE FILTERING

---

**System Prompt**
You are a professional visual data analyst responsible for filtering multi-image sequences for a machine learning dataset. Your task is to ensure that each sequence meets the following strict quality and coherence standards:

1. **Continuity of Action:** The images must depict a single, continuous, and uninterrupted action or process performed by the same subject or agent.

2. **Temporal Coherence:** The sequence must have a clear and logical chronological order. The state change between consecutive frames must be discernible and sensible.

3. **Visual Quality:** All images in the sequence must be clear and free of significant blur, corruption, or distracting artifacts. The main subject and object must be clearly visible.

4. **Scene Consistency:** The background and core environment must remain consistent throughout the sequence. Changes should be due to the action itself, not abrupt scene cuts or significant camera movement.

---

**User Prompt**
Strictly analyze the provided image sequence based on the quality standards. Determine if it represents a high-quality, coherent, and temporally logical process.
A sequence is considered **unqualified** if any of the following are true:

- The images are from different, unrelated scenes or actions.
- The chronological order is illogical or indiscernible.
- The images are severely blurry, low-resolution, or contain duplicate frames.
- The change in the scene is solely due to camera panning/zooming without a meaningful action occurring.
- The main object of the action is swapped, disappears, or is completely occluded.

Does this image sequence meet all the required standards? Please provide your answer as a single word: 'Yes' or 'No'.

---

## C PROMPTS FOR MODEL TRAINING

---

**Reasoning and Output Instruction Template**

You will be presented with a task involving a set of images. The specific task is described in the content below. Carefully analyze the images and the specific task description provided above. Your response must strictly follow the format rules below.

ANSWER FORMAT RULES

Your response format depends on the specific task presented in the content above:

1. **For an Ordering Task:** If the question asks for the correct sequence of images (labeled A, B, C, D), your answer must be a single string representing the correct order of the labels. Do not include spaces or other characters.

2. **For a Multiple-Choice Question (MCQ):** If the question provides options (e.g., A, B, C, D, E), your answer must be the single letter corresponding to the correct option. If you believe none of the options are correct, choose the letter for the "None of the choices provided" option if available.

EXAMPLES

**Example for an Ordering Task:**
`<think>`The process starts with image C, which shows the initial state. Image A adds the first component. Image B continues the process, and image D shows the final, completed assembly. Therefore, the correct sequence is C, then A, then B, then D.`</think>`
`<answer>`CABD`</answer>`

**Example for a Multiple-Choice Question (MCQ):**
`<think>`The question asks to predict the state of the phone screen after tapping the 'Settings' icon. The first image shows the home screen. Option C correctly displays the main settings menu, which is the expected outcome. Options A, B, and D show irrelevant screens.`</think>`
`<answer>`C`</answer>`

You must enclose your reasoning process in `<think>` tags and your final answer in `<answer>` tags. Output only the content within these tags, with no additional text or explanation.

---

## D ABLATION STUDY ON DATA FROM DIFFERENT TRAINING STAGES.

Table 7: Ablation study on the training stage order for the **Qwen2.5-VL-7B** model. All scores are Overall Accuracy (%).

| | Training Strategy | | Benchmark Accuracy (%) | |
| --- | --- | --- | --- | --- |
| *Stage 1* | *Stage 2* | *Stage 3* | **MuirBench** | **LEGO-Puzzles** |
| *Ordering* | *Next Frame Prediction* | *Previous Frame Review* | 61.31 | 39.91 |
| *Ordering* | *Previous Frame Review* | *Next Frame Prediction* | 61.58 | 40.45 |
| *Previous Frame Review* | *Ordering* | *Next Frame Prediction* | 60.38 | 42.64 |
| *Next Frame Prediction* | *Ordering* | *Previous Frame Review* | 64.23 | 42.55 |
| *Next Frame Prediction* | *Previous Frame Review* | *Ordering* | 63.62 | 42.09 |
| *Previous Frame Review* | *Next Frame Prediction* | *Ordering* | 63.62 | 42.09 |
| All tasks combined | | | **65.03** | **42.82** |

## E THE USE OF LLM

During the writing and editing of this paper, the author(s) utilized Large Language Models (such as ChatGPT) for text refinement to improve the clarity and accuracy of the language. These tools

were primarily used for grammar checking, optimizing phrasing, and enhancing readability. All core ideas, the research design, data analysis, and conclusions are the original work of the author(s). The author(s) take full responsibility for the final content of the manuscript and have carefully reviewed all AI-assisted modifications.

# F TRAINING STABILITY AND REWARD ANALYSIS

## F.1 TRAINING STABILITY

To demonstrate the stability of our Reinforcement Learning (RL) training process, we visualize the reward curves in Figure 5. As illustrated, the model's average reward exhibits a rapid and smooth ascent during the initial training stages. Subsequently, the curve successfully converges to a high-score plateau and maintains stability throughout the remaining steps, showing no signs of collapse or drastic oscillations. This convergence trajectory provides strong empirical evidence that our GRPO training configuration is both stable and efficient.

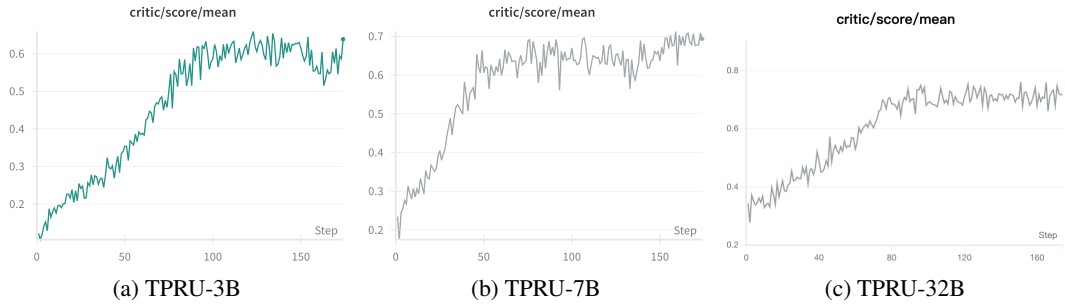

(a) TPRU-3B        (b) TPRU-7B        (c) TPRU-32B

Figure 5: **RL Training Reward Curves.** The plots (a), (b) and (c) display the reward score across training steps. The curves demonstrate rapid initial convergence followed by a stable high-score plateau, indicating a stable optimization process without collapse or drastic oscillations.

## F.2 ABLATION ON FORMAT REWARDS

To investigate whether the inclusion of a format-specific reward induces overfitting to the prompt structure, we conducted a dedicated ablation study. We finetuned the Qwen2.5-VL-7B without the format reward, forcing the model to learn solely from the core task accuracy reward.

As presented in Table 8, while removing the format reward results in a marginal performance decrease, the model retains robust capabilities across all benchmarks. For instance, on the TPRU-test, the performance only drops slightly from 75.70% to 74.40%. These results substantiate that the vast majority of our model's performance stems from the acquisition of core temporal reasoning skills rather than superficial mimicry of the output format. The format reward serves as a beneficial auxiliary mechanism that enhances training stability and provides minor performance gains, but it is not the primary driver of the model's reasoning ability.

Table 8: Ablation study on the impact of format rewards. "TPRU-7B (No-Format Reward)" denotes the model trained solely with task-accuracy rewards, excluding format-specific constraints.

| Model | MuirBench | | LEGO-Puzzles | | TPRU-Test |
|---|---|---|---|---|---|
| | Overall | Ordering | Overall | Ordering | |
| TPRU-7B (No-Format Reward) | 63.96 | 29.69 | 41.64 | 28.0 | 74.40 |
| **TPRU-7B (Ours)** | **65.04** | **34.4** | **42.82** | **32.0** | **75.70** |

