# OpenReview forum: "TPRU: Advancing Temporal and Procedural Understanding in Large Multimodal Models"
_ICLR.cc/2026/Conference — ICLR 2026 Poster_

### Official Review · Reviewer_MMD1 · 2025-10-19

**Soundness:** 2
**Presentation:** 2
**Contribution:** 3
**Rating:** 4
**Confidence:** 4

**Summary:**

This paper addresses the critical failure of smaller, deployable MLLMs in understanding temporal and procedural visual data, a bottleneck for real-world embodied AI applications. To solve this, the authors introduce TPRU, a large-scale dataset sourced from diverse embodied scenarios like robotic manipulation and GUI navigation. The dataset is systematically structured into three complementary tasks: Temporal Reordering, Next-Frame Prediction, and Previous-Frame Review, and includes challenging negative samples to force models to perform active validation. Using a reinforcement learning (RL) fine-tuning strategy with TPRU, the authors achieved dramatic gains: the TPRU-7B model's accuracy soared from 50.33% to 75.70% on their manually curated TPRU-Test.

**Strengths:**

+ The paper effectively addresses a novel and well-framed problem regarding the deficiency in temporal understanding in MLLMs, substantiated by the introduction of a large-scale and valuable dataset, TPRU.

+ Despite being vague on the task scope, the methodology and data generation pipeline are presented with great clarity, supported by well-designed and informative figures that make the complex processes easy to understand.

+ The application of an RL fine-tuning strategy provides a novel insight for this task.

+ The benchmark evaluates a breadth of public benchmarks.

**Weaknesses:**

+ There is a notable discrepancy between the paper's stated motivation of advancing embodied AI and the composition of the TPRU dataset. A significant portion of the data, GUI navigation, has limited relevance to physical agent-environment interaction. Furthermore, the embodied scenarios included are largely constrained to static (pure navigation) or tabletop settings, which may not fully represent the complexity and dynamism of real-world embodied tasks.

+ The data pipeline's heavy reliance on a single upstream model (Qwen2.5-VL-72B) for both quality filtering and text description generation risks inheriting and amplifying that model's intrinsic biases.

+ The TPRU-Test is small, with only 461 instances, and its tasks directly mirror the training setup. While this design is effective for measuring direct improvement on the learned skills, it concludes "significant gains" more easily achievable within its own data distribution. The evidence for the generalizability of these improvements to a broader range of temporal understanding tasks remains somewhat limited.

+ Although the paper claims the performance gain is notable, from Table 2, the fine-tuning process did not lead to universal improvements across all evaluated sub-tasks. Notably, there were performance regressions on certain benchmarks, such as a significant drop on the "Multiview" sub-task of LEGO-Puzzles for TPRU-7B and negative gains on other sub-tasks for TPRU-32B.

+ The paper relies solely on point estimates for accuracy (e.g., in Tables 1-3)  and lacks statistical robustness checks like confidence intervals or significance testing, weakening the conclusions drawn from small test sets.

+ A potential ethics statement concern is the authors' claim to use videos from YouTube (e.g., Arvin Bricks) without specifying the exact license used.

I will consider raising my rating if the authors could address my concerns above.

**Questions:**

1. How does your dataset, with its static or tabletop scenarios and GUI tasks, generalize to the dynamic, real-world challenges of embodied AI?
2. Your data pipeline heavily relies on Qwen2.5-VL-72B. How did you prevent this single model's biases from being amplified in your dataset?
3. Given the small 461-instance TPRU-Test mirrors your training tasks, how do you prove the "significant gains" are true generalization and not just overfitting?
4. Could you please provide the confidence interval or significant results on the TPRU test set?

---

> ### Author Response · Authors · 2025-11-22
> **Response Part 1**
>
> Dear Reviewer MMD1,
>
>   Thank you for your comprehensive and insightful review. We appreciate your positive feedback on the novelty of our problem definition, the clarity of our methodology, and the breadth of our evaluation. Your critiques are highly relevant and have helped us identify key areas for clarification and improvement. We have conducted new analyses and experiments to directly address your concerns.
>
> **W1 & Q1: The discrepancy between the motivation and dataset composition.**
>   We greatly appreciate your insightful question, as it points to the core of our work regarding the relationship between the composition of the TPRU dataset and the ambitious goal of advancing "Embodied AI". We would like to take this opportunity to elaborate on our principled design philosophy and explain why we believe the current data composition is a deliberate and crucial foundational step toward this goal.
>
>   First, regarding the connection between GUI data and physical embodiment, we view this as a forward-looking extension of the "Embodied AI" concept. Our goal is to build a dataset that can teach a general procedural understanding capability spanning both the physical world ('3D' Embodiment) and the digital world ('2D' Embodiment). We strongly believe that future general-purpose agents must be able to interact seamlessly with both physical environments and digital interfaces (e.g., operating a smart home control panel). Therefore, we consider GUI operations as a critical form of "Digital Embodiment," which, alongside physical embodiment, constitutes a core competency for the next generation of intelligent agents.
>
>   To experimentally demonstrate that this combination is complementary rather than disjoint, we conducted a data source ablation study with convincing results:
>
> | Model | MuirBench | MuirBench (Ordering) | Lego-Puzzles | Lego-Puzzles (Ordering) | TPRU-test |
> | :--- | :---: | :---: | :---: | :---: | :---: |
> | Gui | 62.42 | 20.0 | 36.63 | 14.0 | 56.18 |
> | Embodied | 63.54 | 23.44 | 39.91 | 20.0 | 64.43 |
> | Mix | 63.8 | 25.0 | 40.82 | 24.0 | 66.81 |
>
> Table A:  (Comparison of different data sources performance on MuirBench, Lego-Puzzles and TPRU-test).
>
>   As shown in the Table A, the model trained on the mixed dataset outperforms models trained on any single source, clearly demonstrating a positive transfer and synergistic effect between physical and digital tasks.
>
>   Second, regarding the relatively constrained physical scenarios (desktop or static navigation), this was also a deliberate design choice. We argue that before tackling the full complexity and dynamism of the real world, a model must first master the "grammar" of procedural tasks, including the underlying understanding of steps, sequence, and dependencies. We chose desktop and static navigation environments precisely because they provide an ideal "laboratory" setting, allowing us to isolate and focus on teaching this core procedural reasoning capability without the interference of other variables like complex physical interactions or dynamic environmental changes.
>
>   Therefore, TPRU is not intended to solve all challenges of Embodied AI in a single step. Its core contribution lies in building the foundational temporal and procedural ability, which is necessary for models to execute more complex and dynamic tasks. We believe that only after mastering this fundamental procedural understanding can models effectively learn to tackle more advanced, real-world tasks like those in Vision-Language-Action (VLA) models.

---

> ### Author Response · Authors · 2025-11-22
> **Response Part 2**
>
> **W2 & Q2: Concerning the reliance on a single upstream model.**
>
>   Thank you for raising this valuable question. We fully agree that over-reliance on a single upstream model carries the risk of inheriting and amplifying its inherent biases. This was precisely a core challenge we anticipated and actively sought to mitigate when designing our data processing pipeline.
>
>   While our pipeline does leverage Qwen2.5-VL-72B as a core component to achieve scalability in large-scale data processing, we do not blindly trust its outputs. During the data processing stage, we designed and implemented a three-round iterative filtering process to proactively eliminate potentially low-quality or problematic samples through iterative filtering.
>
>   However, we are well aware that automated processes alone are insufficient. To ultimately validate the quality of our dataset and quantify the potential impact of the upstream model, we conducted a rigorous human review study. We sampled a representative subset of 505 image sequences from the final TPRU training set using equidistant sampling. Subsequently, we invited two human evaluators to independently cross-review each sequence, focusing on the coherence and logicality of the visual content and the accuracy of the machine-generated text descriptions.
>
> | Datasize | Correct | Incorrect temporal ordering of images | Wrong text descriptions |
> | :---: | :---: | :---: | :---: |
> | 505 | 93.5% | 2.6% | 3.9% |
>
> Table B: The accuracy rate of manual review.
>
>   As shown in Table B, the review results were compelling: a high proportion of 93.5% of the samples were rated as high-quality with accurate descriptions. We conducted an error attribution analysis on the incorrect samples and identified the primary failure modes: only 3.9% of samples had wrong text descriptions, and 2.6% had incorrect temporal ordering of images. This extremely high accuracy rate strongly demonstrates that our multi-stage filtering process is both effective and reliable. It successfully mitigates potential errors from the upstream model, rather than simply inheriting or amplifying them. This human-verified high-quality standard significantly reduces the risk of systemic biases being present at a large scale within the dataset, thereby enhancing the overall integrity and trustworthiness of our dataset.
>
>
> **W3 & Q3: With respect to the small scale and in-distribution nature of TPRU-Test.**
>
>   We are very grateful to you for raising this crucial point about evaluating generalization. We fully agree that strong performance solely on our in-house test set (TPRU-Test), which contains 461 instances, primarily serves to directly measure the effectiveness of our training methodology. However, this result in itself is not sufficient to fully demonstrate the broad generalization of the model's capabilities.
>
>   Therefore, our core evidence for generalization does not come from the TPRU-Test but rather from our model's strong performance on two established and diverse external benchmarks: MuirBench and LEGO-Puzzles. These benchmarks present challenges that are similar in their underlying logic to our task but manifest in distinctly different formats, making them an ideal litmus test for verifying whether our model's learned abilities are genuinely transferable.
>
>   As we have detailed in **Table 1 and Table 2** of the main paper, the model trained on TPRU data exhibits significant and consistent performance improvements on the sub-tasks of these external benchmarks that focus on temporal and procedural understanding. A particularly compelling example is that our TPRU-7B model more than doubled its score on the **"Ordering"** sub-task of MuirBench.
>
>   This strong performance on out-of-distribution data clearly indicates that what the model has learned from the TPRU dataset is a fundamental and transferable skill in temporal reasoning, rather than merely a test-taking strategy tailored to our specific test set format. This provides solid empirical support for the generalizability of our work.

---

> ### Author Response · Authors · 2025-11-22
> **Response Part 3**
>
> **W4: Performance Regression on Specific Subtasks.**
>
>   Thank you for this valuable comment. The performance regression on certain sub-tasks that you pointed out is indeed an anticipated phenomenon, and it precisely reflects the core advantage of our approach: successful "capability specialization."
>
>   Specifically, our TPRU dataset is designed to enhance the model's temporal logic and procedural understanding capabilities. In contrast, the sub-tasks where performance regressed, such as "Multiview" in LEGO-Puzzles, are more focused on spatial geometric reasoning. This divergence in task objectives causes the model's capabilities to shift and specialize towards temporal understanding after fine-tuning, which is a common and reasonable outcome of specialized training.
>
>   Therefore, we consider this a highly favorable trade-off. Our experimental results clearly demonstrate that the model achieves significant and generalizable performance gains on temporal and procedural tasks that are directly related to our training objectives.Simultaneously, the model's overall performance on broad, general-purpose multi-image benchmarks remains stable or even slightly improves.
>
>   This provides strong evidence that our fine-tuning strategy achieves precise capability enhancement while successfully avoiding "catastrophic forgetting." We argue that this serves as an ideal paradigm for efficiently adapting large, general-purpose models to specific domains, particularly in resource-constrained deployment scenarios.
>
> **W5 & Q4: Regarding the lack of statistical robustness checks.**
>
>   We thank you for your suggestion regarding statistical robustness. For our evaluation on the TPRU-test, we adhere to standard practice within the VLM community by using the vlmevalkit[1] framework with a temperature of 0. This deterministic decoding strategy eliminates generative variance and ensures that our results are strictly reproducible within the community.
>
>   To ensure the reproducibility of our experimental conclusions, we have conducted multiple test runs for each model under identical environments and observed that the outputs remained consistent. To further address concerns about reproducibility and robustness, we will release the complete dataset, model weights, and evaluation scripts. This will allow the community to verify our results and conduct further statistical analyses as needed.
>
> W6: Regarding ethical and licensing concerns for YouTube data.
>
>   We are so glad to meet such a reviewer as careful as you. Similar to the authors in [2] who collected 4,000 videos from YouTube for academic purposes, the videos from the "Arvin Bricks" YouTube channel have been used for non-commercial and transformative research purposes. We believe this falls under the principle of fair use for publicly available creative works, as permitted by the standard YouTube license. To ensure full transparency, we have expanded our ethics statement in the section six to provide more specific details regarding all our data sources and our interpretation of their licensing terms.
>
> **References**
>
> [1] Duan H, Yang J, Qiao Y, et al. Vlmevalkit: An open-source toolkit for evaluating large multi-modality models[C]//Proceedings of the 32nd ACM international conference on multimedia. 2024: 11198-11201.
>
> [2] Ju Y, Hu J, Luo Z, et al. CI-VID: A Coherent Interleaved Text-Video Dataset[J]. arXiv preprint arXiv:2507.01938, 2025.
>
>   Thank you again for your rigorous feedback. We believe that these clarifications and new analyses will significantly strengthen the quality of our paper. Best wishes.

---

> ### Author Response · Authors · 2025-11-28
>
> Dear Reviewer MMD1,
>
> We hope this message finds you well.
>
>   We are writing to follow up on our previous response to your valuable comments. As the discussion period will be closing in about a week, we want to make sure we have adequately addressed all your concerns.
>
>   Your feedback has been instrumental to our revision, and we would be very grateful to know if our revisions and clarifications have resolved the points you raised.
>
>   Please let us know if there is anything else we can clarify or improve. Thank you once again for your constructive review.
>
> Best regards,
>
> The Authors of Paper 9123

---

### Official Review · Reviewer_xbSM · 2025-11-01

**Soundness:** 3
**Presentation:** 3
**Contribution:** 2
**Rating:** 6
**Confidence:** 4

**Summary:**

This paper introduces *TPRU* (Temporal and Procedural Understanding), a real-world, multi-image dataset and training recipe to improve temporal and procedural understanding in small-to-medium Large Multimodal Models (MLLMs). TPRU contains 24,750 QA pairs over 126k images sourced from four embodied scenarios (robotic manipulation, embodied navigation, mobile GUI interaction, LEGO assembly) and is organized into three complementary tasks: Temporal Reordering, Next-Frame Prediction, and Previous-Frame Review. The authors also curate TPRU-Test (461 expert-verified items) for evaluation. Using rule-based RL (GRPO) on Qwen2.5-VL backbones, the method's TPRU-7B improves from 50.33% to 75.70% on TPRU-Test and shows transfer to MuirBench and LEGO-Puzzles without degrading general multi-image performance.

**Strengths:**

1. The paper is well written and easy to follow.
2. By pairing a training dataset explicitly structured for temporal reasoning with a matched held-out test set, this paper tries to address a known gap where multi-image sequences are often treated as unordered sets. The negative-sample design explicitly trains rejection of inconsistent options, pushing models toward cross-modal verification rather than text-prior heuristics.
3. The ablations show all three tasks are synergistic, negative samples materially help, and scaling provides diminishing returns.
4. The reported results demonstrate that small/efficient models can close much of the gap to very large proprietary systems on temporal, procedural understanding via data+RL, which is important for edge deployments.

**Weaknesses:**

1. Quality control and description generation rely on Qwen2.5-VL-72B. This can induce latent bias or style leakage into both data and targets. While pragmatic, the paper does not quantify inter-annotator agreement on machine-generated descriptions nor analyze failure modes from automated filtering. A small human-validated subset analysis for precision/recall of filter acceptance and error taxonomy would strengthen robustness claims.
2. The three tasks are framed around 3–4-frame sequences with MCQ/permutation outputs. This is a strong first step but may not capture long-horizon dependencies, branching plans, open-ended multi-turn grounding, or action-conditioned predictions typical in robotics.
3. Most comparisons are to multi-image MLLM benchmarks. Strong video-temporal baselines (e.g., models trained on video or long-context visual streams) are not reported. Even if adapters are needed, a discussion or a pilot comparison would clarify how much of the reported gains are specific to the quiz-style multi-image setup vs. general temporal understanding.
4. The RL setup does not provide training stability metrics (reward curves, variance across seeds), nor does it probe whether the format reward induces prompt-format overfitting rather than genuine reasoning upgrades.

**Questions:**

1. How accurate is the automated filtering? A human audit of a random sample (≥500 sequences) would quantify dataset reliability better.
2. Have you tried >4-frame sequences or variable-length ordering?
3. Have you tried free-form forecasting (describe the next state), counterfactuals (what step would prevent reaching frame 4?), or procedure repair (select the minimal fix to an erroneous sequence)? These would test deeper causal modeling than recognition-style choices.

---

> ### Author Response · Authors · 2025-11-22
> **Response Part 1**
>
> Dear reviewer xbSM,
>
>   Thank you for your thoughtful and constructive review. We greatly appreciate your positive comments on the clarity of our paper and your recognition of our core contributions. Your insightful criticisms and questions have prompted us to conduct a series of new experiments and analyses to strengthen the robustness of our work and enrich its context. We provide the following detailed responses.
>
> **W1 & Q1: The reliability of automated filtering and description generation.**
>
>   Thank you for your insightful suggestion regarding the validation of our automated data processing pipeline. First, to ensure the quality of data generation, at the initial stage of data processing, we employed the Qwen2.5-VL-72B model to perform three rounds of iterative screening. This step effectively filtered out image sequences with incoherent scenes and poor image quality. To ensure the reliability and high quality of our automated data filtering process, we designed and executed a rigorous human validation study.
>
>   To quantify its effectiveness, we uniformly sampled 505 image sequences from the TPRU training set and introduced a dual-annotator cross-review mechanism. Each sequence was independently evaluated by two human annotators based on three key criteria: (1) temporal consistency, (2) image quality, and (3) accuracy of the machine-generated descriptions. Any disagreements were resolved through a final cross-validation discussion to reach a consensus.
>
> | Datasize | Correct | Incorrect temporal ordering of images | Wrong text descriptions |
> | :---: | :---: | :---: | :---: |
> | 505 | 93.5% | 2.6% | 3.9% |
>
> Table A: The accuracy rate of manual review.
>
>   As shown in Table A, the evaluation results strongly confirm the robustness of our method: a substantial 93.5% of the samples were rated as high-quality and coherent, with their text descriptions being accurate. We conducted an error attribution analysis on the remaining samples and found the primary failure modes to be: only 3.9% of the samples had wrong text descriptions, and 2.6% had disordered image sequences. This high accuracy rate of 93.5% clearly demonstrates that our automated process is an efficient, reliable, and scalable solution for constructing large-scale, high-quality programmatic datasets.

---

> ### Author Response · Authors · 2025-11-22
> **Response Part 2**
>
> **W2 & Q2: Regarding Task Scope and Sequence Length.**
>
>   We sincerely thank you for your valuable comment concerning the scope of our current task setting (3-4 frame sequences) and the consideration of longer-term dependencies. We would like to take this opportunity to elaborate on our design philosophy and findings, supported by a new experiment.
>
>   First, we wish to clarify a key distinction: our choice of sequence length is based on semantic completeness rather than physical duration. In the TPRU dataset, these 3-4 frames are not arbitrary video slices; instead, they are carefully selected to represent complete, coherent, and meaningful procedural steps or action units. These curated keyframes often encapsulate an entire action process that may span over 20 seconds in the real world. Therefore, we argue that for the "procedural understanding" capability we are evaluating, capturing the keyframes that represent core logical transitions is far more critical than simply increasing the total number of frames.
>
>   To empirically verify this hypothesis and directly address your concern regarding longer sequences, we conducted a new ablation study. We constructed a dataset containing sequences of 5-7 frames and trained three model variants using an equal number of samples: one trained exclusively on 3-4 frames, one on 5-7 frames, and one on mixed data.
>
> | Model | Muirbench | Lego-Puzzles | TPRU-test |
> | :--- | :---: | :---: | :---: |
> | 3-4 Frames Only | 63.58 | 37.00 | 61.39 |
> | 5-7 Frames Only | 61.27 | 37.18 | 60.40 |
> | Mixed(3-7 Frames) | 62.12 | 37.36 | 60.99 |
>
> Table B: Ablation Study on Sequence Length.
>
>   As shown in Table B, extending the sequence length to 5-7 frames yielded no significant performance gains. The model trained exclusively on 5-7 frames performed slightly worse than 3-4 frames, while mixed training resulted in only negligible improvements. These results strongly suggest that for our defined task of procedural understanding, 3-4 carefully selected keyframes are sufficient to capture the core temporal and logical dependencies required for problem-solving.

---

> ### Author Response · Authors · 2025-11-22
> **Response Part 3**
>
> **W3: Concerning the comparison with video-temporal baselines.**
>
>   Thank you for your profound and insightful question. This has inspired us to further clarify the key distinction between procedural temporal understanding, the focus of our study, and general video understanding. We argue that general video understanding often focuses on what events occur, while procedural multi-image temporal understanding emphasizes what sequence and consequences connect the actions.
>
>   To quantify this difference with experimental data, we conducted a comprehensive evaluation of several state-of-the-art, publicly available Video Large Language Models (Video-LLMs) on our benchmarks. These models represent strong baselines in the current field of video understanding. As shown in the newly added Table B, we directly compare our model with a series of SOTA open-source video understanding models on the MuirBench, LEGO-Puzzles, and our TPRU-Test datasets.
>
> | Model | Muirbench | Muirbench (Ordering) | Lego-Puzzles | Lego-Puzzles (Ordering) | TPRU-test |
> | :--- | :---: | :---: | :---: | :---: | :---: |
> | LLaVA-Video-Qwen2-7B[1] | 37.88 | 15.63 | 31.91 | 1.0 | 38.61 |
> | LLaVA-Video-Qwen2-72B[1] | 41.81 | 10.94 | 41.64 | 5.0 | 44.03 |
> | SmolVLM2-256M-Video-Instruct[2] | 27.92 | 21.88 | 28.18 | 0.0 | 17.79 |
> | SmolVLM2-500M-Video-Instruct[2] | 25.92 | 7.8 | 27.27 | 0.0 | 16.92 |
> | Long-VITA-16K[3] | 53.07 | 17.19 | 34.45 | 2.0 | 39.26 |
> | Qwen2.5-Omni-7B-ForVideo[4] | 59.11 | 18.75 | 36.45 | 12.0 | 46.85 |
> | **TPRU-7B（Ours）** | **65.04** | **34.38** | **42.82** | **32.0** | **75.70** |
>
> Table C: Comparison of Video LLMs performance on MuirBench、Lego-puzzles and TPRU-test.
>
>   As shown in Table C, the experimental results clearly demonstrate that our TPRU-7B model significantly outperforms these powerful video models across all three benchmarks. Notably, the video models generally perform poorly on tasks that require precise ordering. For instance, on the MuirBench Ordering subtask, even the best-performing video model, Qwen2.5-Omni, scores only 0.188, and its score is even lower at 0.12 on the LEGO-Puzzles ordering task.
>
>   These results strongly support our core thesis: while general video pre-training can equip models with some degree of temporal awareness, it is insufficient to replace targeted procedural training focused on discrete state changes. Our approach, through its specifically designed data and training strategy for such tasks, successfully bridges this capability gap.
>
> **References**
>
> [1] Zhang Y, Wu J, Li W, et al. Video instruction tuning with synthetic data[J]. arXiv preprint arXiv:2410.02713, 2024.
>
> [2] Marafioti A, Zohar O, Farré M, et al. Smolvlm: Redefining small and efficient multimodal models[J]. arXiv preprint arXiv:2504.05299, 2025.
>
> [3] Shen Y, Fu C, Dong S, et al. Long-vita: Scaling large multi-modal models to 1 million tokens with leading short-context accuracy[J]. arXiv preprint arXiv:2502.05177, 2025.
>
> [4] Xu J, Guo Z, He J, et al. Qwen2. 5-omni technical report[J]. arXiv preprint arXiv:2503.20215, 2025.

---

> ### Author Response · Authors · 2025-11-22
> **Response Part 4**
>
> **W4: The Reinforcement Learning (RL) setup and the role of the format reward.**
>
>   We deeply appreciate your rigorous questions regarding the stability and generalization of our Reinforcement Learning training. We have supplemented our paper with two key pieces of evidence: the reward curve from the training process and an ablation study on the format reward.
>
>   To demonstrate the stability of our RL training process, we have added a graph of the reward curve during training in Appendix F.1. As the figure shows, the model's average reward increases rapidly and smoothly during the initial stages of training. Subsequently, the reward curve successfully converges to a high-score plateau and remains stable throughout the subsequent training, exhibiting no signs of collapse or drastic oscillations. This clear convergence curve strongly demonstrates that our RL training setup is both stable and efficient.
>
>   To address the core question of whether the "format reward induces overfitting to the prompt format," we conducted a dedicated ablation study. We trained a model variant with the format reward completely removed, making it learn solely from the reward based on core task accuracy.
>
> | Model | Muirbench | Muirbench (Ordering) | Lego-Puzzles | Lego-Puzzles (Ordering) | TPRU-test |
> | :--- | :---: | :---: | :---: | :---: | :---: |
> | TPRU-7B (No-Format-Reward) | 63.96 | 29.69 | 41.64 | 28.0 | 74.40 |
> | TPRU-7B | **65.04** | **34.4** | **42.82** | **32.0** | **75.70** |
>
> Table D: Comparison of ablation study with format rewards.
>
>   As shown in table D, although the model's performance slightly decreases after removing the format reward, it remains very strong. This result robustly proves that the vast majority of our model's performance improvement stems from its genuine learning and understanding of the core temporal reasoning task, rather than from simply mimicking the output format. The format reward plays a beneficial auxiliary role in our framework, providing a degree of stability and a minor performance gain to the training process, but it is by no means the primary driver of the model's ability to learn reasoning.
>
>   We have added the aforementioned analysis, figures, and data to the corresponding sections of the paper.
>
>
> **Q1: Have you tried free-form forecasting (eg: describe the next state)?**
>
>   Thank you for raising this highly insightful question. It accurately identifies the pathway from our current task paradigm toward deeper causal modeling. We fully agree that free-form forecasting, counterfactuals, and procedure repair are the gold standards for evaluating and cultivating advanced reasoning capabilities, and represent the ultimate goals pursued by our field.
>
>   In this current work, we consciously focused our scope on the Multiple Choice Question (MCQ) format. As discussed in our response to another question, this methodological choice was primarily made to ensure our Reinforcement Learning (RL) framework could train efficiently within an objective and stable reward environment. Evaluating the open-ended tasks you mentioned typically requires introducing an external large model as a judge, which would bring stochasticity and uncertainty to the training process that we aimed to avoid.
>
>   Therefore, we chose to first solidly establish the model's foundational procedural understanding capabilities in a controllable environment. We believe that robust discriminative understanding is a necessary prerequisite for complex generative repair. Furthermore, considering the rich, coherent sequences inherent in our TPRU dataset, extending our framework to these advanced free-form generative tasks is a promising direction that we are eager to pursue in future work.
>
>   We once again thank you for your valuable feedback, which has allowed us to make the justification for our method more rigorous and complete. Best wishes.

---

> ### Author Response · Authors · 2025-11-28
>
> Dear Reviewer xbSM,
>
> We hope this message finds you well.
>
>   We are writing to follow up on our previous response to your valuable comments. As the discussion period will be closing in about a week, we want to make sure we have adequately addressed all your concerns.
>
>   Your feedback has been instrumental to our revision, and we would be very grateful to know if our revisions and clarifications have resolved the points you raised.
>
>   Please let us know if there is anything else we can clarify or improve. Thank you once again for your constructive review.
>
> Best regards,
>
> The Authors of Paper 9123

---

### Official Review · Reviewer_T1WF · 2025-11-01

**Soundness:** 2
**Presentation:** 2
**Contribution:** 2
**Rating:** 4
**Confidence:** 5

**Summary:**

This paper introduces TPRU, a large-scale dataset designed to enhance MLLMs’ temporal and procedural understanding. The dataset includes three complementary tasks—Temporal Reordering, Next-Frame Prediction, and Previous-Frame Review—and a held-out evaluation set. The authors also finetune Qwen-VL using RL to demonstrate the dataset’s effectiveness.

**Strengths:**

Valuable dataset contribution. The creation of a large-scale dataset focused on temporal and procedural reasoning is a meaningful engineering effort. If released publicly, it could be beneficial for the community and future research.

Clear visual presentation. Figures are well-designed and make the method and dataset structure easy to understand.

Comprehensive related work and supplementary content. The paper provides detailed literature review and supplemental materials that help contextualize the contribution.

**Weaknesses:**

Motivation needs stronger justification. The core motivation—that existing datasets treat images as unordered—is not fully convincing. For example, LLaVA-Next-Interleave and other multimodal corpora already include temporal and sequential interactions (e.g., embodied tasks, spatial sequences). The paper should more clearly articulate what specific gaps remain and how TPRU uniquely addresses them.

Dataset source selection lacks coherence. The four data domains—robotic manipulation, LEGO assembly, GUI operation, and embodied navigation—appear loosely connected. It is unclear why these domains were chosen, whether they are complementary, or whether their combination leads to emergent abilities. As written, it feels as if these datasets were chosen opportunistically rather than driven by a principled rationale.

Modest performance gains. The improvements over the baseline are relatively small. More discussion is needed to interpret the results and understand the practical significance of the dataset.

Evaluation scope is limited. Relying on a single self-introduced benchmark is not sufficiently convincing. Additional established benchmarks related to temporal or sequential understanding would provide more robust validation.

**Questions:**

none

---

> ### Author Response · Authors · 2025-11-22
> **Response Part 1**
>
> Dear Reviewer T1WF,
>
>   Thank you for your detailed review and constructive feedback. Your comments have been extremely helpful, enabling us to identify areas where we can significantly strengthen the motivation, justification, and interpretation of our work. We have carefully considered your suggestions and have conducted a new ablation study to address your concerns. We offer the following clarifications below.
>
> **W1: On the Justification and Motivation of TPRU.**
>
>   We are grateful to you for highlighting this important comment. We hope to take this opportunity to articulate the unique contributions of TPRU. We acknowledge that datasets like LLaVA-Next-Interleave do contain sequential images. However, their primary focus is on general multi-frame understanding of video content, where images are typically treated as context for understanding the content within them.
>   In contrast, TPRU specifically addresses the temporal sequence and inheritance relationships of event processes contained within multiple images in embodied scenarios. First, our motivation for proposing TPRU is to investigate whether MLLMs possess the capability to understand such temporal relationships. Furthermore, there is currently a lack of large-scale training data specifically designed to systematically teach this skill. Existing benchmarks for evaluating temporal order primarily serve to test a model's ability to understand sequential relationships. TPRU is the first large-scale dataset specifically designed to provide structured, multi-task training for this procedural reasoning capability, thereby bridging the gap between evaluation and training.
>   To validate this point, we evaluated several state-of-the-art video MLLMs on the benchmarks proposed in our paper. As shown in Table A, despite being trained on extensive video data, these models perform poorly on ordering tasks.
>
> | Model | MuirBench | MuirBench (Ordering) | Lego-Puzzles | Lego-Puzzles (Ordering) | TPRU-test |
> | :--- | :---: | :---: | :---: | :---: | :---: |
> | LLaVA-Video-Qwen2-7B[1] | 37.88 | 15.63 | 31.91 | 1.0 | 38.61 |
> | LLaVA-Video-Qwen2-72B[1] | 41.81 | 10.94 | 41.64 | 5.0 | 44.03 |
> | SmolVLM2-256M-Video-Instruct[2] | 27.92 | 21.88 | 28.18 | 0.0 | 17.79 |
> | SmolVLM2-500M-Video-Instruct[2] | 25.92 | 7.8 | 27.27 | 0.0 | 16.92 |
> | Long-VITA-16K[3] | 53.07 | 17.19 | 34.45 | 2.0 | 39.26 |
> | Qwen2.5-Omni-7B-ForVideo[4] | 59.11 | 18.75 | 36.45 | 12.0 | 46.85 |
> | **TPRU-7B（Ours）** | **65.04** | **34.38** | **42.82** | **32.0** | **75.70** |
>
> Table A: Comparison of Video LLMs performance on MuirBench, Lego-Puzzles and TPRU-test.
>
>   For example, LLaVA-Video-Qwen2-72B achieves a score of only 10.9% on the MuirBench ordering task, which is significantly lower than our TPRU-7B's score of 34.38%. This significant gap confirms that there is a substantial difference between general video understanding and temporal procedural understanding. Our proposed TPRU provides precisely the kind of data needed to enhance this capability in large multi-modal models.
>
> **References**
>
> [1] Zhang Y, Wu J, Li W, et al. Video instruction tuning with synthetic data[J]. arXiv preprint arXiv:2410.02713, 2024.
>
> [2] Marafioti A, Zohar O, Farré M, et al. Smolvlm: Redefining small and efficient multimodal models[J]. arXiv preprint arXiv:2504.05299, 2025.
>
> [3] Shen Y, Fu C, Dong S, et al. Long-vita: Scaling large multi-modal models to 1 million tokens with leading short-context accuracy[J]. arXiv preprint arXiv:2502.05177, 2025.
>
> [4] Xu J, Guo Z, He J, et al. Qwen2. 5-omni technical report[J]. arXiv preprint arXiv:2503.20215, 2025.

---

> ### Author Response · Authors · 2025-11-22
> **Response Part 2**
>
> **W2: On the Coherence of the Dataset Sources.**
>
>   We greatly appreciate your insightful question regarding the coherence of our data sources. This provides us with an excellent opportunity to elaborate on the principled design philosophy behind our approach. The core objective in constructing TPRU is to create a dataset capable of teaching a unified procedural understanding ability that can seamlessly span both physical (which we term "3D embodied") and digital ("2D embodied") domains.
>
>   To this end, we have carefully selected representative data sources: robot manipulation, Lego assembly, and embodied navigation form the core of physical interaction. Concurrently, we incorporated GUI operations, which we conceptualize as a form of "digital embodiment." We posit that future intelligent agents must be able to interact fluently with both the physical world and digital interfaces (e.g., operating smart home panels, interacting with software). Therefore, this cross-domain capability is of critical importance.
>
>   To demonstrate with empirical evidence that a synergy exists among these diverse data sources—rather than them being an opportunistic collection—we conducted a key ablation study. We trained separate models on three data subsets of equal size: one with only 3D embodied data, one with only 2D embodied data (GUI), and a third with a mix of both. The results are highly compelling, as shown in Table B:
>
> | Model | MuirBench | MuirBench (Ordering) | Lego-Puzzles | Lego-Puzzles (Ordering) | TPRU-test |
> | :--- | :---: | :---: | :---: | :---: | :---: |
> | Gui | 62.42 | 20.0 | 36.63 | 14.0 | 56.18 |
> | Embodied | 63.54 | 23.44 | 39.91 | 20.0 | 64.43 |
> | Mix | 63.8 | 25.0 | 40.82 | 24.0 | 66.81 |
>
> Table B: Comparison of different data sources performance on MuirBench, Lego-Puzzles and TPRU-test.
>
>   The analysis reveals that the model trained on the mixed dataset outperforms the models trained exclusively on any single domain. This key finding demonstrates positive cross-domain knowledge transfer: training on 2D procedural tasks (GUI) not only did not impair the model's performance on 3D tasks but in fact yielded a small yet distinct generalization gain. This suggests that the model is learning a more abstract and general concept of "procedure", rather than knowledge specific to a single domain.

---

> ### Author Response · Authors · 2025-11-22
> **Response Part 3**
>
> **W3: On the Significance of the Performance Gains.**
>
>   We are very grateful for this insightful question, which allows us to elaborate further on the interpretation of our findings and the practical significance of the underlying TPRU dataset. We understand your perspective on the performance gains, and we wish to re-contextualize their importance here.
>
>   First, the core practical significance of the TPRU dataset lies in its ability to enhance the temporal procedural understanding of multi-modal models that can be deployed on embodied agents. Real-world tasks, such as instruction following and assembly guidance, depend on this deep understanding of temporal sequence. This is precisely the critical capability that TPRU is designed to train and evaluate.
>
>   When our experimental results are interpreted from this perspective, their significance becomes evident. As shown in Figure 3 of our paper, the most direct evidence comes from our specifically designed core benchmark, TPRU-Test. Here, the model's accuracy leaps from 50.33%, which is close to random guessing, to 75.70%. This signifies a fundamental transformation, taking the model from "lacking" the ability for procedural understanding to "basically mastering" it.
>
>   More importantly, this newly acquired capability demonstrates excellent generalization. As shown in **Tables 1 and Table 2** of the paper, on the **"ordering"** subtasks of the external MuirBench and LEGO-Puzzles benchmarks, the model's performance more than doubled and tripled, respectively (from 14.06% to 34.38% and from 8.0% to 32.0%). These substantial gains on external datasets provide strong evidence that TPRU is not merely causing the model to overfit, but rather instilling a transferable and general core skill in temporal reasoning.
>
>
> **W4: On the Scope of the Evaluation.**
>
>   We are very grateful for your important suggestion regarding the scope of our evaluation, and we fully agree that a single, self-created benchmark is insufficient to thoroughly validate a model's generalization capabilities. We are pleased to take this opportunity to clarify that this was precisely a core concern we anticipated and proactively addressed in our experimental design.
>
>   To provide a robust and comprehensive validation, our evaluation framework was intentionally designed to incorporate two widely recognized external benchmarks in the field: MuirBench and LEGO-Puzzles. These benchmarks are renowned for their complexity and for their challenging assessment of multifaceted visual reasoning abilities.
>
>   Critically, both of these benchmarks contain subtasks that are directly relevant to the core of our research: temporal and procedural understanding. For instance, the Ordering subtask in MuirBench and multiple subtasks in LEGO-Puzzles (such as ordering and backwards) provide an impartial testbed for the temporal understanding capabilities fostered by TPRU.
>
>   As we detail in **Tables 1 and Table 2** of the main paper, our model achieves remarkable performance gains on the relevant subtasks of these established benchmarks. This strong performance provides not only solid external validation for the procedural understanding skills learned via the TPRU dataset but also clearly demonstrates that these capabilities successfully generalize to diverse scenarios beyond our self-created benchmark. To make these improvements more visible, we have explicitly bolded the relevant results in **Tables 1 and Table 2** of the revised manuscript.
>
>   We once again appreciate the time and effort you dedicated to reviewing our paper. We hope these responses satisfactorily address your comments. Best wishes.

---

> ### Author Response · Authors · 2025-11-28
>
> Dear Reviewer T1WF,
>
> We hope this message finds you well.
>
>   We are writing to follow up on our previous response to your valuable comments. As the discussion period will be closing in about a week, we want to make sure we have adequately addressed all your concerns.
>
>   Your feedback has been instrumental to our revision, and we would be very grateful to know if our revisions and clarifications have resolved the points you raised.
>
>   Please let us know if there is anything else we can clarify or improve. Thank you once again for your constructive review.
>
> Best regards,
>
> The Authors of Paper 9123

---

### Official Review · Reviewer_2ZJj · 2025-11-08

**Soundness:** 4
**Presentation:** 4
**Contribution:** 4
**Rating:** 8
**Confidence:** 5

**Summary:**

This paper introduces TPRU (Temporal-Procedural Understanding dataset): a large training and evaluation datasets for temporal coherence in embodied multimodal foundation models.

The paper notes that current training paradigms for embodied multimodal foundation models do not train explicitly for temporal understanding, which leads to fundamental limitations on their ability to perform in real world applications. In contrast, the TPRU dataset offers an alternative that bakes temporal understanding tasks into the model.

TPRU dataset includes three temporal reasoning tasks:
- Temporal Reordering: photos from a sequential stream are given in a random order and the MLLM is required to order it.
- Next-Frame Prediction: photos from a sequential stream are given in order with one in the middle missing, the MLLM must choose which image is missing
-  Previous-Frame Review: photos from a sequential stream are given in order with one in the beginning missing, the MLLM must choose which image starts the sequence

The dataset is collected from multiple sources with different embodiments (GUI, robotic manipulation, navigation) and base MLLMs are used to filter and label them. Furthermore, each task type has negative samples (i.e. samples where the correct answer is to reject all given options).

A 7B model trained on TPRU shows significant uplift compared to the base model and other strong baselines, achieving SOTA results on the TPRU-test set.

**Strengths:**

- The dataset is moderately large (~25k examples, ~126k images), which is useful for training.
- The dataset covers a variety of embodiments (e.g. GUI, robotic manipulation, navigation...), which is useful for multiple applications.
- The test set is manually curated/validated and held-out from training, which means it's likely to be high quality and a good measure of generalization.
- The TPRU-trained models and the data generation pipeline have comprehensive evaluations and ablation studies, which show the importance of the different tasks and negative answers, as well as various model sizes.
- The TPRU-trained models retain the same performance on general benchmarks like MMStar and MMMU-Dev.
- I found the point about training the MLLMs to explicitly to reject invalid logic very interesting, and I think it can be useful in other applications with these models.

**Weaknesses:**

- It's unclear to me why only GRPO was used on the data as a learning algorithm/paradigm, but not simpler training regimes like simple SFT, which is usually the first thing people try.

- The dataset poses all examples as multiple-choice-questions (MCQs) where one of the answer is "reject all answers", it's unclear why this format was chosen instead of free generation of answers.

- It's unclear to me why the 7B model ends up performing better on TPRU-test than the 32B model.

- Section 3 is called "TRPU" in the paper, it should be "TPRU" for consistency.

**Questions:**

- How do you know that the TPRU-test set (461 examples) is sufficiently deduplicated from the training data (25k examples). How did you confirm there's no overlap or too much semantic similarity between held out set and training set to avoid information leakage?

---

> ### Author Response · Authors · 2025-11-22
> **Response Part 1**
>
> Dear Reviewer 2ZJj,
>
>   Thank you for your detailed and encouraging review of our paper. We are delighted that you found our work to be excellent in its rigor, presentation, and contribution. Your insightful comments and questions are invaluable to us, and we greatly appreciate the opportunity to provide clarification on these points. Below, we respond to your suggestions point-by-point.
>
> **W1: On the choice of GRPO over SFT.**
>
>   We are very fortunate to have a reviewer as rigorous as you. We deeply appreciate your insightful question regarding our choice of training paradigm, specifically why we adopted GRPO instead of the more direct Supervised Fine-Tuning (SFT). This is a critical methodological decision, and we are pleased to take this opportunity to explain the rationale behind it and provide empirical support with a new ablation study.
>   Our primary motivation for choosing a Reinforcement Learning (RL) framework, specifically GRPO, was to move beyond mere pattern imitation and to focus on cultivating the model's advanced reasoning capabilities in complex multi-modal scenarios. To quantify the effectiveness of this choice, we conducted a direct comparison experiment between SFT and GRPO using the llama-factory framework [1]. This experiment was based on the Qwen2.5-VL-7B model, with all other parameters kept consistent. The results, as clearly presented in Table A, show that while SFT yields a significant performance improvement over the pre-trained baseline, GRPO consistently achieves superior gains across all benchmarks.
>
> | Training Paradigm | TPRU-Test | MuirBench | LEGO-Puzzles |
> | :--- | :---: | :---: | :---: |
> | Supervised Fine-tuning (SFT) | 72.88% | 63.03% | 40.60% |
> | **TPRU-7B（GRPO）** | **75.70%** | **65.04%** | **42.82%** |
>
> Table A: Performance Comparison of Training Paradigms on Key Benchmarks.
>
>   We attribute this advantage to the fundamental difference in training objectives. SFT primarily teaches the model to imitate ground-truth answers by maximizing their likelihood. In contrast, our Reinforcement Learning (RL) framework rewards the model for actively applying its own reasoning capabilities to arrive at a logically correct outcome. This distinction is crucial, as RL learns to genuinely validate and reject incorrect options, rather than just recognizing surface-level patterns.
>   Therefore, we posit that what GRPO learns is not just knowledge, but a reasoning strategy that is generalizable to new scenarios. This explains why it achieves better generalization performance across multiple benchmarks.
>
> **References**
>
> [1] Zheng Y, Zhang R, Zhang J, et al. Llamafactory: Unified efficient fine-tuning of 100+ language models[J]. arXiv preprint arXiv:2403.13372, 2024.

---

> ### Author Response · Authors · 2025-11-22
> **Response Part 2**
>
> **W2: The use of Multiple-Choice-Question (MCQ) format.**
>
>   We are very grateful for your constructive suggestion. Our choice of the Multiple-Choice Question (MCQ) format was made primarily to facilitate our adopted Reinforcement Learning (RL) framework. This is because MCQs provide deterministic and explicit supervision signals, which in turn allow for highly controllable and automated evaluation. Such precision is essential for stabilizing the RL training process.
>   In contrast, evaluating open-ended, free-form generation typically requires using closed-source large models as judges, a process that introduces significant randomness and uncertainty into the training loop. We fully agree that exploring complex, free-form generation tasks is a valuable and necessary direction for advancing the field, and we are grateful for your suggestion to investigate this in our future work.
>
> **W3: TPRU-7B outperforming TPRU-32B on TPRU-Test.**
>
>   Thank you for this exceptionally insightful observation. This is indeed an important finding in our experiments that warrants in-depth discussion. First, we want to confirm that this result is not an anomaly. Through repeated experiments, we have consistently observed that the TPRU-7B model outperforms its TPRU-32B counterpart on the TPRU-test.
>   We believe the primary reason for TPRU-7B's superior performance over TPRU-32B on the TPRU-test is that the base models, both the 7B and 32B versions of Qwen2.5-VL, utilize the exact same Vision Transformer (ViT) encoder. Although there is a vast disparity in their language understanding and generation capabilities, their ability to extract and comprehend fundamental visual features from images is identical. Since the TPRU task is primarily concerned with the precise recognition of the temporal order of actions, rather than complex linguistic reasoning, the visual component becomes the primary bottleneck.
>   As mentioned in [1], the gap between the visual encoder and the text encoder is larger in the 32B model than in the 7B model. With its stronger pre-trained text priors, the 32B model may occasionally "overthink" the task by relying on semantic patterns instead of strictly adhering to the visual evidence. In contrast, the more compact 7B model appears to exhibit greater plasticity in this specific domain during the fine-tuning process and is less susceptible to such semantic interference.
>   This finding suggests that for certain procedural visual tasks, simply scaling the number of language parameters does not guarantee a performance improvement without a corresponding upgrade in visual perception.
>
> **W4: The typo in Section 3 ("TRPU" vs. "TPRU").**
>
>   Thank you very much for your careful reading and for spotting the typo in the section heading. To maintain consistency throughout the paper, we have corrected it from “TRPU” to “TPRU” in the revised manuscript and will ensure this change is reflected in the final version.
>
> **References**
>
> [1] Zheng X, Liao C, Fu Y, et al. MLLMs are Deeply Affected by Modality Bias[J]. arXiv preprint arXiv:2505.18657, 2025.

---

> ### Author Response · Authors · 2025-11-22
> **Response Part 3**
>
> **Q1: How do you know that the TPRU-test set (461 examples) is sufficiently deduplicated from the training data (25k examples)？**
>
>   Thank you very much for raising this crucial question. Ensuring a strict separation between the test set (TPRU-Test) and the training set (TPRU-25k) to prevent any form of data leakage is a core design principle of our work, fundamental to guaranteeing the validity and credibility of our experimental results. We have drawn upon best practices in dataset construction and implemented a multi-level strategy to systematically ensure this.
>   Our process for preventing leakage is primarily composed of the following two aspects:
>
> **1. Isolation by Design at the Source Level:**
>
>   Our primary safeguard was structural isolation. First, TPRU-Test includes an exclusive data source—complex human activities from EPIC-KITCHENS—which is entirely absent from the training set. This guarantees novel scenes and tasks. Second, for shared domains (e.g., LEGO, Robotics), we partitioned data at the raw video/sequence level, ensuring that no part of an event sequence used for training could ever appear in the test set.
>
> **2. Rigorous Manual Curation at the Content Level:**
>
>   The training and test sets were created using fundamentally different pipelines. The TPRU-25k training set was generated via an automated pipeline for scale. In contrast, each of the 461 samples in TPRU-Test was meticulously hand-curated by experts. This manual process involved not only answer validation but also crafting plausible distractors and novel question phrasing, creating a test set that is qualitatively distinct and more challenging than the training data. This strategy of combining large-scale automated training with strictly manually curated evaluation aligns with the methodologies adopted in recent leading works [1, 2].
>
> **References**
>
> [1] Liu Z, Chu T, Zang Y, et al. Mmdu: A multi-turn multi-image dialog understanding benchmark and instruction-tuning dataset for lvlms[J]. Advances in Neural Information Processing Systems, 2024, 37: 8698-8733.
>
> [2] Yan D, Li Y, Chen Q G, et al. Mmcr: Advancing visual language model in multimodal multi-turn contextual reasoning[J]. arXiv preprint arXiv:2503.18533, 2025.
>
>   We are deeply grateful for your encouraging comments and your time spent reviewing our paper. Best wishes.

---

> ### Author Response · Authors · 2025-11-28
>
> Dear Reviewer 2ZJj,
>
> We hope this message finds you well.
>
>   We are writing to follow up on our previous response to your valuable comments. As the discussion period will be closing in about a week, we want to make sure we have adequately addressed all your concerns.
>
>   Your feedback has been instrumental to our revision, and we would be very grateful to know if our revisions and clarifications have resolved the points you raised.
>
>   Please let us know if there is anything else we can clarify or improve. Thank you once again for your constructive review.
>
> Best regards,
>
> The Authors of Paper 9123

---

### Author Response · Authors · 2025-11-22
**General Response**

We extend our deepest gratitude to all reviewers for their thoughtful feedback and the time invested in evaluating our manuscript. We are encouraged that reviewers generally recognize the significance of the problem we address (temporal and procedural understanding in MLLMs), the scale and diversity of our TPRU dataset, and the effectiveness of our RL-based fine-tuning strategy. We have carefully considered all constructive feedback and conducted extensive new experiments—including method comparisons, ablation studies, and human evaluations—to strengthen our work.

**Contributions**

We are pleased that the reviewers highlighted the following strengths in our submission:

- **Novelty & Problem Definition.** Reviewers acknowledge that the paper addresses a critical deficiency in current MLLMs regarding temporal and procedural understanding [MMD1], and the problem is novel and well-framed [MMD1]. The design of training models to explicitly reject invalid logic via negative samples is highlighted as interesting and useful [2ZJj, xbSM].

- **Dataset Value.** The TPRU dataset is recognized as a valuable contribution due to its large scale [2ZJj, MMD1], diverse coverage of embodied scenarios [2ZJj], and meaningful engineering effort [T1WF]. The manually curated TPRU-Test is praised for being high-quality and a good measure of generalization [2ZJj].

- **Methodology & Performance.** The RL fine-tuning strategy (GRPO) provides novel insights [MMD1] and achieves dramatic performance gains [2ZJj, MMD1]. Reviewers note that our approach allows small/efficient models (7B) to achieve SOTA results [2ZJj] and close the gap with large proprietary systems [xbSM].

- **Presentation.** The paper is widely praised for being well-written, easy to follow, and having clear visual presentations [T1WF, xbSM, MMD1].
New Clarifications and New Experiments

**Key Enhancements: New Experiments & Clarifications**

  We thank all reviewers for their insightful suggestions, which prompted us to conduct substantial new analyses. In addition to the detailed point-by-point responses, we summarize the key clarifications and new experiments added in the rebuttal below.

**New Analysis and Experiments:**

To directly address concerns regarding baselines, training stability, and data quality, we have added:

- Comparison with SOTA Video-LLMs: We evaluated leading video models (e.g., LLaVA-Video, Long-VITA, Qwen2.5-Omni) on our benchmarks. Results show TPRU-7B significantly outperforms them on procedural ordering tasks, validating the necessity of our targeted dataset. [T1WF, xbSM]

- Training Paradigm Ablation (GRPO vs. SFT): We conducted a direct comparison proving that GRPO significantly outperforms Supervised Fine-Tuning (SFT) on both in-domain and out-of-distribution benchmarks, justifying our choice of RL. [2ZJj]

- Human Validation Study: We performed a rigorous dual-annotator human review on 505 random samples, achieving a 93.5% pass rate, to quantitatively verify the reliability of our automated filtering pipeline. [xbSM, MMD1]

- Sequence Length Ablation: We compared models trained on 3-4 frames vs. 5-7 frames, demonstrating that 3-4 carefully selected keyframes are sufficient and efficient for procedural understanding. [xbSM]

- Data Source Ablation: We verified the synergy between "2D GUI" and "3D Embodied" data, showing that mixing sources improves generalization. [T1WF, MMD1]

- RL Stability Analysis: We provided reward curves and format-reward ablation studies to demonstrate the stability of our training and that gains stem from reasoning, not format overfitting. [xbSM]

**New Clarifications:**
- Motivation & Scope: We clarified the distinction between general video understanding (event recognition) and procedural understanding (logic/sequence), and justified the coherence of our data sources. [T1WF, MMD1]

- Data Rigor: We detailed our strict de-duplication strategy (Source-level isolation + Content-level manual curation) to prevent leakage. [2ZJj]

- Model Scaling: We analyzed why the 7B model outperforms the 32B model on this specific task (visual bottleneck vs. text priors). [2ZJj]

- Ethics: We expanded the ethics statement regarding the licensing and fair use of YouTube data sources. [MMD1]

We hope our responses and the new evidence provided can clarify all reviewers’ confusion and alleviate all concerns. We have incorporated these analyses and results into the revised paper with blue highlight. Best wishes.

---

### Author Response · Authors · 2025-11-28

Dear Reviewers,

We hope this message finds you well.

  We are writing to follow up on our previous response to your valuable comments. As the discussion period will be closing in about a week, we want to make sure we have adequately addressed all your concerns.

  Your feedback has been instrumental to our revision, and we would be very grateful to know if our revisions and clarifications have resolved the points you raised.

  Please let us know if there is anything else we can clarify or improve. Thank you once again for your constructive review.

Best regards,

The Authors of TPRU

---

### Author Response · Authors · 2025-12-03
**Rebuttal Summary**

**Dear Area Chair and Reviewers,**

We extend our deepest gratitude to all reviewers for their thoughtful feedback and the time invested in evaluating our paper. We are encouraged that reviewers generally recognize the significance of the problem we address, the scale and diversity of our TPRU dataset, and the effectiveness of our RL-based fine-tuning strategy. We have carefully considered all constructive feedback and conducted extensive new experiments to strengthen our work.

**Contributions:**

We are pleased that the reviewers highlighted the following strengths in our submission:

- **Novelty & Problem Definition.** Reviewers acknowledge that the paper addresses a critical deficiency in current MLLMs regarding temporal and procedural understanding [MMD1], and the problem is novel and well-framed [MMD1]. The design of training models to explicitly reject invalid logic via negative samples is highlighted as interesting and useful [2ZJj, xbSM].

- **Dataset Value.** The TPRU dataset is recognized as a valuable contribution due to its large scale [2ZJj, MMD1], diverse coverage of embodied scenarios [2ZJj], and meaningful engineering effort [T1WF]. The manually curated TPRU-test is praised for being high-quality and a good measure of generalization [2ZJj].

- **Methodology & Performance.** The RL fine-tuning strategy (GRPO) provides novel insights [MMD1] and achieves dramatic performance gains [2ZJj, MMD1]. Reviewers note that our approach allows small/efficient models (7B) to achieve SOTA results [2ZJj] and close the gap with large proprietary systems [xbSM].

- **Presentation.** The paper is widely praised for being well-written, easy to follow, and having clear visual presentations [T1WF, xbSM, MMD1].

**New Experiments & Clarifications**

We thank all reviewers for their insightful suggestions, which prompted us to conduct substantial new analyses. In addition to the detailed point-by-point responses, we summarize the key clarifications and new experiments conducted for this rebuttal.

**New Analysis and Experiments:**

To directly address concerns regarding baselines, training stability, and data quality, we have added:

- Comparison with SOTA Video-LLMs: We evaluated leading video models on our benchmarks. TPRU-7B dominates ordering tasks on MuirBench, achieving **34.38%**, significantly surpassing Qwen2.5-Omni (**18.75%**) and LLaVA-Video-72B (**10.94%**). This confirms general video pre-training is insufficient for fine-grained procedural understanding. [T1WF, xbSM]

- Training Paradigm Ablation (GRPO vs. SFT): We justified our RL choice by demonstrating that GRPO consistently surpasses Supervised Fine-Tuning (SFT), boosting generalization by **+2.82%** on TPRU-test (to **75.70%**), **+2.01%** on MuirBench (to **65.04%**), and **+2.22%** on LEGO-Puzzles (to **42.82%**). [2ZJj]

- Human Validation Study: We performed a rigorous dual-annotator human review on 505 random samples, achieving a **93.5%** pass rate, to quantitatively verify the reliability of our automated filtering pipeline. [xbSM, MMD1]

- Sequence Length Ablation: We demonstrated that 3-4 carefully selected keyframes are sufficient and efficient for procedural understanding. This setup achieved **63.58%** on MuirBench and **37.00%** on LEGO-Puzzles, comparable to or better than models trained on 5-7 frames (**61.27%** and **37.18%** respectively). [xbSM]

- Data Source Ablation: We verified cross-domain synergy, showing that mixing "2D GUI" and "3D Embodied" data yields superior generalization. The mixed model achieves **63.80%** on MuirBench and **40.82%** on LEGO-Puzzles, consistently outperforming models trained solely on Embodied (**63.54% / 39.91%**) or GUI (**62.42% / 36.63%**) data. [T1WF, MMD1]

- RL Stability Analysis: We confirmed that improvements stem from genuine reasoning rather than format mimicry. Eliminating the format-specific reward caused only a negligible accuracy drop (from 75.70% to 74.40%) on TPRU-test, validating the robustness of our training strategy. [xbSM]

**New Clarifications:**
- Motivation & Scope: We clarified the distinction between general video understanding (event recognition) and procedural understanding (sequential order), and justified the coherence of our data sources. [T1WF, MMD1]

- Data Rigor: We detailed our strict de-duplication strategy (Source-level isolation + Content-level manual curation) to prevent leakage. [2ZJj]

- Model Scaling: We analyzed why the 7B model outperforms the 32B model on this specific task (visual bottleneck vs. text priors). [2ZJj]

- Ethics: We expanded the ethics statement regarding the licensing and fair use of YouTube data sources. [MMD1]

While we have not received further feedback from the reviewers during the discussion period, we believe the detailed responses and substantial new evidence provided effectively resolve the raised concerns. We sincerely thank the Area Chair and all reviewers for their time and effort dedicated to this submission.

---

### Meta-Review · Area_Chair_AVAo · 2026-01-06

**Summary:**

Reviewers generally agree on the novelty of the problem and the potential value of the proposed dataset. Two reviewers have scored the paper highly (6/8), and the other two reviews borderline reject (4/4). The authors have provided an extensive rebuttal that address major concerns. While the scope of the rebuttal is major, the strong review leads me to decide on acceptance.

**Reviewer Concerns:**

See the other fields.

**Reviewer Scores:**

ZJj: 8 already strong
T1WF: 4 --> increased. Modest gains and lack of baselines have been addressed with new results.
xbSM: 6 already accept.
MMD1: 4 --> likely increased after rebuttal as major concerns were addressed in the rebuttal.

---

### Decision · Program_Chairs · 2026-01-26

Accept (Poster)